# A novel microglial subset plays a key role in myelinogenesis in developing brain

Agnieszka Wlodarczyk[1], Inge R Holtman[2], Martin Krueger[3], Nir Yogev[4,†], Julia Bruttger[4], Reza Khorooshi[1], Anouk Benmamar-Badel[1,5], Jelkje J de Boer-Bergsma[6], Nellie A Martin[7], Khalad Karram[4], Isabella Kramer[1], Erik WGM Boddeke[2], Ari Waisman[4] (iD), Bart JL Eggen[2] & Trevor Owens[1,*] (iD)

## Abstract

Microglia are resident macrophages of the central nervous system that contribute to homeostasis and neuroinflammation. Although known to play an important role in brain development, their exact function has not been fully described. Here, we show that in contrast to healthy adult and inflammation-activated cells, neonatal microglia show a unique myelinogenic and neurogenic phenotype. A CD11c[+] microglial subset that predominates in primary myelinating areas of the developing brain expresses genes for neuronal and glial survival, migration, and differentiation. These cells are the major source of insulin-like growth factor 1, and its selective depletion from CD11c[+] microglia leads to impairment of primary myelination. CD11c-targeted toxin regimens induced a selective transcriptional response in neonates, distinct from adult microglia. CD11c[+] microglia are also found in clusters of repopulating microglia after experimental ablation and in neuroinflammation in adult mice, but despite some similarities, they do not recapitulate neonatal microglial characteristics. We therefore identify a unique phenotype of neonatal microglia that deliver signals necessary for myelination and neurogenesis.

Keywords CD11c; IGF1; microglia; myelinogenesis
Subject Categories Immunology; Neuroscience
The EMBO Journal (2017) 36: 3292–3308

See also: **ML Bennett & BA Barres** (November 2017)

## Introduction

Microglia are resident myeloid cells of the central nervous system (CNS) that originate from yolk sac precursors and colonize the brain very early during embryonic life (Ginhoux *et al*, 2010; Schulz *et al*, 2012; Kierdorf *et al*, 2013). They are autonomously maintained through proliferation (Askew *et al*, 2017) and are not replaced from blood-derived precursors during the lifetime of the host, at least under normal circumstances (reviewed in Ginhoux *et al*, 2013; Prinz & Priller, 2014). The role of microglia has traditionally been studied in the context of immune responses in the diseased CNS, and they have been implicated in neuroinflammatory and neurodegenerative diseases. Although their importance in homeostasis and brain development is recognized (Biber *et al*, 2014; Matcovitch-Natan *et al*, 2016), their exact role in these processes remains incompletely defined. Evidence supporting the view that microglia are crucial players in neurodevelopment includes that mice lacking the CSF1R receptor for CSF1 or IL34, which is critical for microglia maintenance, show abnormal brain development (Elmore *et al*, 2014). Moreover, they are involved in synaptic pruning (Paolicelli *et al*, 2011; Schafer *et al*, 2012; Kettenmann *et al*, 2013; Zhan *et al*, 2014), they modulate axonal outgrowth and cortical interneuron positioning (Squarzoni *et al*, 2014), and they support survival of layer V cortical neurons during postnatal development by producing neuroprotective insulin-like growth factor 1 (IGF1) (Ueno *et al*, 2013). IGF1 has also been shown to be essential for primary myelination, but the microglial contribution to this process is relatively unstudied.

In many neuroinflammatory (Wlodarczyk *et al*, 2014, 2015) and degenerative (Butovsky *et al*, 2006) conditions, a normally rare subpopulation of microglia that expresses the integrin complement receptor CD11c increases in proportion and number. We have previously shown in an animal model for multiple sclerosis, experimental autoimmune encephalomyelitis (EAE), that while CD11c[+] microglia are effective antigen-presenting cells for T-cell proliferation, they are a poor source of pro-inflammatory cytokines (Wlodarczyk *et al*, 2014) and that they differ from infiltrating DC and CD11c[−] microglia with respect to expression of many genes (Wlodarczyk *et al*, 2015). Interestingly, we showed that CD11c[+] microglia uniquely express

1 Department of Neurobiology Research, Institute for Molecular Medicine, University of Southern Denmark, Odense, Denmark
2 Department of Neuroscience, Medical Physiology, University Medical Center Groningen, University of Groningen, Groningen, The Netherlands
3 Institute for Anatomy, University of Leipzig, Leipzig, Germany
4 Institute for Molecular Medicine, University Medical Center of the Johannes Gutenberg University of Mainz, Mainz, Germany
5 Department of Biology, Ecole Normale Supérieure de Lyon, University of Lyon, Lyon, France
6 Department of Genetics, University Medical Center Groningen, University of Groningen, Groningen, The Netherlands
7 Department of Neurology, Institute of Clinical Research, Odense University Hospital, Odense, Denmark
*Corresponding author. Tel: +45 65503951; E-mail: towens@health.sdu.dk
†Present address: Department of Neurology, University Medical Center of the Johannes Gutenberg University of Mainz, Mainz, Germany

*Igf1* during EAE (Wlodarczyk *et al*, 2015), suggesting that they may be neuroprotective.

Here, we show that neonatal microglia differ dramatically in their gene expression profile from microglia in healthy adults and mice with EAE, showing a neurogenic signature. CD11c$^+$ microglia greatly expand during postnatal development (PN3-5) and then dramatically contract as mice age to adulthood. They express a characteristic neurosupportive gene profile, equipping them to play a fundamental role in the developing CNS. Moreover, they are appropriately located for delivery of signals necessary for neuronal development and primary myelination. Importantly, we show that *Igf1* deficiency in CD11c$^+$ microglia leads to an impairment in primary myelination. Three separate CD11c-targeted toxin regimens all resulted in a similar selective transcriptional outcome in neonatal mice, with a response distinct from that of adult microglia. Furthermore, whereas CD11c$^+$ microglia are among the cells that repopulate the adult brain after microglial ablation, they do not show myelinogenic or neurogenic signatures. Thus, we identify neonatal microglia, and especially the CD11c$^-$ subset, as key tissue macrophages for CNS development.

# Results

## CD11c$^+$ microglia emerge during postnatal development

During neuroinflammation, CD11c$^+$ microglia are the major source of *Igf1* (Wlodarczyk *et al*, 2015), a gene critical for neurodevelopment, myelination, and neurogenesis. Thus, we asked whether this microglial subset is present during postnatal development. Mononuclear cells were isolated from perfused brains from PN2, PN3, PN7, PN28, and adult (8–12 weeks old) B6 and CCR2$^{rfp/+}$ mice and analyzed by flow cytometry for CD11c expression. Microglia are defined by a lower level of cell surface CD45 than blood-derived leukocytes, expression of fractalkine receptor CX3CR1, and lack of CCR2 chemokine receptor (Mizutani *et al*, 2012). We used relative CD45 levels (Remington *et al*, 2007) and CCR2 expression to discriminate between blood-derived leukocytes (CD45$^{high}$ CCR2$^+$) and resident microglia (CD45$^{dim}$CCR2$^-$). There were very few CD45$^{high}$ CD11c$^+$ cells (0.2% live gate) in brain isolates at indicated time points. Nevertheless, nearly 85% of them were CCR2-positive (Fig 1A). In contrast, more than 99% of CD45$^{dim}$ CD11b$^+$ CD11c$^+$ cells (microglia) were CCR2-negative (Fig 1A). The proportion of CD11c$^+$ microglia, as a percentage of total microglia, increased significantly from PN2 (12%) to PN3 (17%) and then sharply decreased by PN7 (8%), falling to < 3% in young (PN28) and adult animals (Fig 1B). Absolute numbers of CD11c$^+$ microglia significantly increased from PN2 to PN3-5. After PN5, the numbers of CD11c$^+$ microglia dramatically decreased, reaching only 50 and 10% of PN3-5 levels at PN7 and PN28, respectively (Fig 1C). Numbers of CD11c$^-$ microglia were also elevated in neonatal CNS, increasing from PN2 to PN3. However, unlike CD11c$^+$ microglia, their numbers were stable from PN3 throughout adulthood (Fig 1D). Immunofluorescent stainings of perfused PN4-5 B6 and CX3CR1$^{GFP/+}$ murine brains showed that CD11c$^+$ microglia cells were not homogenously distributed throughout the brain, but localized mainly in the corpus callosum and cerebellar white matter, in contrast to other areas of the brain where they were virtually absent (Fig 1E–G). All of the

CD11c-positive cells co-stained with the microglial marker IBA1 (Fig 1H) and co-localized with CX3CR1 (Fig 1I). Importantly, laminin staining revealed that they were localized in the brain parenchyma and not in blood vessels or in the perivascular space (Fig 1J).

## Neonatal CD11c$^+$ microglia are a critical source of IGF1 for primary myelination

Next, we were interested whether CD11c$^+$ microglia, so abundant in areas of primary myelination, are a source of *Igf1* that is critical for this process. We compared levels of *Igf1* expression in MACS-sorted neurons, astrocytes, oligodendrocyte precursor cells (OPCs), and CD11b$^+$ cells (which include mostly microglia) from unperfused PN4-7 brains. Although all cell populations expressed detectable levels of *Igf1*, CD11b$^+$ cells showed at least eightfold higher expression of this gene (Fig 2A). *Igf1* transcripts were further compared in CD11c$^+$ and CD11c$^-$ microglial populations. CD11c$^+$ microglia expressed significantly higher levels of *Igf1* (sevenfold) than their CD11c$^-$ counterparts (Fig 2B).

To confirm the importance of CD11c$^+$ microglia-derived IGF1 on primary myelination, we used a CD11c-Cre-GFP driver to delete the *Igf1 gene* specifically in CD11c$^+$ cells. In line with data presented in Fig 1, Cre-GFP positive cells were localized mainly in corpus callosum (Fig 2C) and cerebellum. We confirmed that all of these cells co-stained with CD11b (Fig 2C). PCR analysis of genomic DNA in sorted OPC, astrocytes, neurons, and microglia from CD11c$^{Cre-GFP}$ Igf1$^{fl/fl}$ PN7 mice revealed Cre-induced recombination only in microglia (Fig 2D). We additionally showed lack of recombination in astrocytes from PN21 CD11c$^{Cre-GFP}$ Igf1$^{fl/fl}$ mice, and we showed recombination in microglia but not in neurons, OPC, or astrocytes from PN7 CD11c$^{Cre-GFP}$ Igf1$^{fl/WT}$ heterozygous mice (Fig EV1). This is in line with results from Goldmann *et al* (2013), who showed that in a CD11c$^{Cre}$: Rosa EYFP reporter mouse EYFP was exclusively expressed by Iba1 + microglia in CNS parenchyma and that there was no ectopic expression outside the myeloid lineage in these mice. Flow cytometry analysis showed that CD11c$^+$ but not CD11c$^-$ microglia or CD45 negative cells were Cre-GFP positive (not shown). *Igf1* expression was significantly reduced in sorted Cre$^+$ CD11c$^+$ microglia, reaching levels of CD11c$^-$ microglia and splenic CD11c$^+$ cells (Fig 2E). Even though the efficiency of Cre recombination in CD11c$^+$ microglia was only close to 40% (Fig 2D), we observed lower brain weight (Fig 2F), significant decrease in *Igf1*, *Plp*, *Mag*, and *Mbp*, and slight downregulation of *Mog* gene expression (Fig 2G) in PN21 CD11c$^{Cre-GFP}$ Igf1$^{fl/fl}$ brains in comparison with *Igf1*-intact Igf1$^{fl/fl}$ littermate controls. This was accompanied by less intense PLP staining (Fig 2H) and significantly higher myelin *G*-ratio (Fig 2I–K) in corpus callosum. Conversely, we observed higher representation of less myelinated fibers (*G*-ratio 0.8–0.85) and significantly less frequent sufficiently myelinated axons (*G*-ratio = 0.65–0.7) in corpus callosum of CD11c$^{Cre-GFP}$ Igf1$^{fl/fl}$ than in littermate controls (Fig 2L and M). Altogether, our data point to an important role for IGF1-producing CD11c$^+$ microglia in primary myelination.

## Distinct gene signatures in microglia subsets during development and EAE

We have previously shown that numbers and proportions of CD11c$^+$ microglia dramatically increase during EAE, and the data

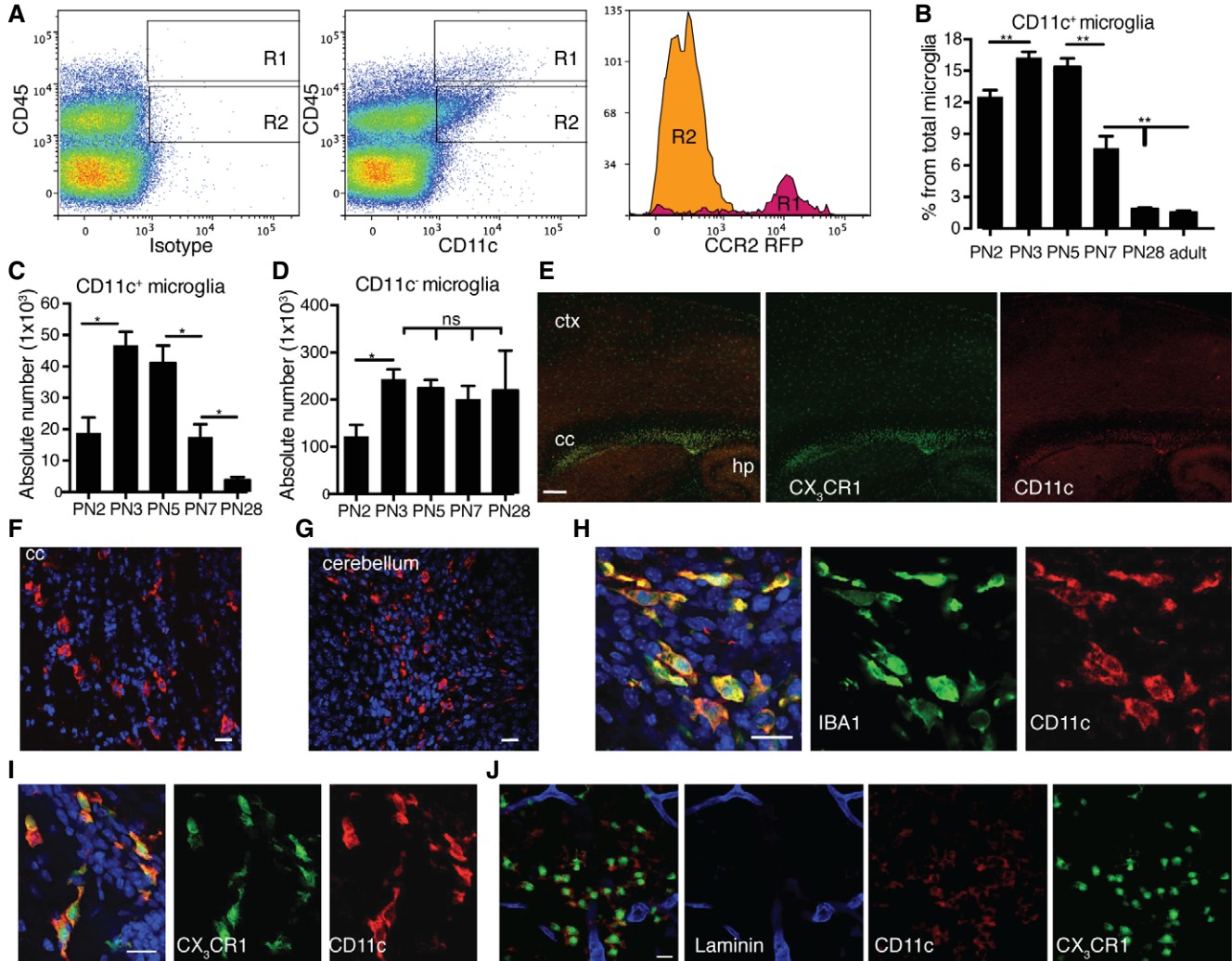

**Figure 1. CD11c⁺ microglia emerge during postnatal neurodevelopment.**

A    Representative flow cytometry profiles of five individual brain suspensions prepared from PN5 mice showing RFP expression driven by *Ccr2* promoter on CD45^high^CD11c⁺ cells (R1) and lack of expression on CD45^low^CD11c⁺ microglia (R2).

B–D  Flow cytometry analysis showing CD11c⁺ microglia presented as a percentage from total microglia (B), absolute numbers of CD11c⁺ microglia (C), and CD11c⁻ microglia (D) at different time points PN2 (*n* = 4), PN3 (*n* = 6), PN5 (*n* = 6), PN7 (*n* = 6), PN28 (*n* = 9), and adult mouse brain (*n* = 16).

E    Representative low-power micrographs showing patches of CD11c-stained cells (red) co-localizing with GFP driven by *Cx3cr1* promoter (green) in corpus callosum (cc) and single CX3CR1-positive cells in cortex (ctx) and hippocampus (hp) in PN4-5 brains (*n* = 3). Scale bar = 200 μm.

F–I  Representative confocal microscopic micrographs showing CD11c-stained cells (red) in corpus callosum (F) and cerebellum (G) as well as co-localization of CD11c marker (red) and IBA1 (green) (H) or CX3CR1 (green) (I) in PN4-5 brains (*n* = 3). Scale bars = 15 μm.

J    Confocal microscopic analysis of two individual brains showing CD11c (red), CX3CR1 (green) double-positive cells localized in the parenchyma, outside of the laminin-stained blood vessels (blue) (*n* = 2). Scale bar = 15 μm.

Data information: Data are based on three experimental repeats. Data are presented as means ± SEM; each *n* represents an individual mouse. *P*-values were determined by two-tailed Mann-Whitney *U*-test. ns, not significant; **P* < 0.05; ***P* < 0.01.

suggested that they might play a protective role during the disease (Wlodarczyk *et al*, 2014, 2015). To assess whether microglial subsets from developing CNS and mice with symptomatic EAE are similar, we compared transcriptomes of sorted CD11c⁺ and CD11c⁻ microglia from PN4-6 CNS and symptomatic EAE (grades 3-5) as well as of total naïve adult microglia (Dataset EV1). Microglial markers (*Aif1, Itgam, Cx3cr1, Csf1r*) and signature genes (Butovsky *et al*, 2014; Bennett *et al*, 2016) (*Spi1, Irf8, Olfml3, Hexb, Fcrls, Tgfbr1, P2ry12, Siglech, Tmem119*) were

similarly expressed in both CD11c⁺ and CD11c⁻ neonatal and EAE microglia populations as in adult microglia (not shown). CD11c⁺ microglia from neonates and mice with EAE strongly upregulated *Itgax* expression, confirming high purity of sorted cells and validating the RNA-seq assay (not shown). Moreover, we identified 20 genes (*Itgax, Gpnmb, Spp1, Igf1, Colec12, Ccl5, Ak4, Lox, Mmp12, Cpeb1, Ntn1, Clec7a, Saa3, Ahnak2, Fabp5, Hpse, Gm26902, Cspg4, Fam20c*, and *Stra6 l*) that were associated with the CD11c⁺ microglia population.

    

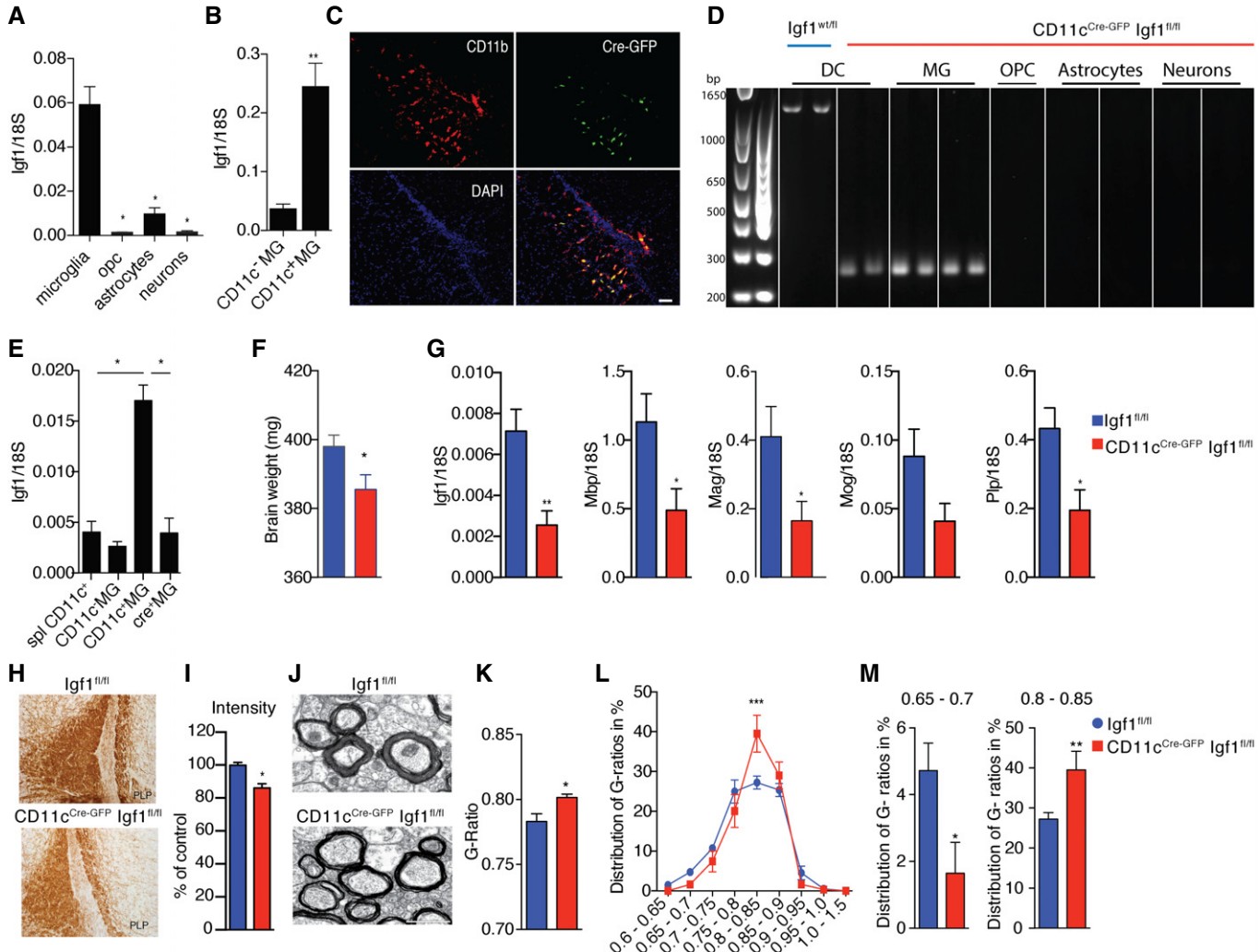

**Figure 2.  Neonatal CD11c⁺ microglia are a major source of myelinogenic *Igf1*.**

A, B    Expression of *Igf1* relative to 18S rRNA in MACS-sorted microglia, OPC, astrocytes and neurons (*n* = 4) (A) as well as FACS-sorted CD11c⁺ and CD11c⁻ microglia (*n* = 6) (B) from brains of PN4-7 mice.

C    Representative micrographs showing patches of Cre-GFP, CD11b double-positive cells in corpus callosum from PN4-5 CD11c ᶜʳᵉ⁻ᴳᶠᴾ Igf1ᶠˡ/ᶠˡ brains (*n* = 3), Scale bar = 50 μm.

D    Genomic PCR analysis of Cre recombination in MACS-sorted splenic dendritic cells (DC) from Igf1ʷᵗ/ᶠˡ and microglia, OPC, astrocytes, and neurons from CD11cᶜʳᵉ⁻ᴳᶠᴾ Igf1ᶠˡ/ᶠˡ. Wild-type *Igf1* gene is detected as an ~1-kb band; Cre-induced recombination is detected as an ~0.2-kb band, while the Igf1/flox locus cannot be amplified under the assay condition (Liu *et al*, 1998).

E    Expression of *Igf1* relative to 18S rRNA in FACS-sorted splenic CD11c⁺ cells (*n* = 5), CD11c⁺ microglia (*n* = 5), and CD11c⁻ microglia (*n* = 5) from Igf1ᶠˡ/ᶠˡ mice and Cre⁺ microglia (*n* = 4, each *n* represents a pool of 2 brains) from CD11cᶜʳᵉ⁻ᴳᶠᴾ Igf1ᶠˡ/ᶠˡ mice.

F    Bar graph showing weights of brains from PN21 Igf1ᶠˡ/ᶠˡ (blue) (*n* = 8) and CD11cᶜʳᵉ⁻ᴳᶠᴾ Igf1ᶠˡ/ᶠˡ (red) (*n* = 4) mice.

G    Expression of *Igf1*, *Mog*, *Plp*, and *Mbp* relative to 18S rRNA in brain tissue from PN21 CD11cᶜʳᵉ⁻ᴳᶠᴾ Igf1ᶠˡ/ᶠˡ and Igf1ᶠˡ/ᶠˡ mice (F) *n* = 6.

H, I    Representative micrographs (H) and quantification of PLP staining intensity (I) in corpus callosum of CD11cᶜʳᵉ⁻ᴳᶠᴾ Igf1ᶠˡ/ᶠˡ (red) (*n* = 4) and Igf1ᶠˡ/ᶠˡ (blue) (*n* = 6) PN21 brains.

J–M    Representative electron microscopy micrographs (J), mean G-ratios (K), and distribution of G-ratios (L, M) in corpus callosum from Igf1ᶠˡ/ᶠˡ (blue) (*n* = 8) and CD11c CD11cᶜʳᵉ⁻ᴳᶠᴾ Igf1ᶠˡ/ᶠˡ (red) (*n* = 6) PN21 brains. Scale bar = 1 μm.

Data information: Data are based on at least two experimental repeats. Data are presented as means ± SEM; each *n* represents an individual mouse. *P*-values were determined by two-tailed Mann-Whitney *U*-test (A, B, E, G, I), Welch's *t*-test (F, K, M) (distribution was normal) or two-way ANOVA with Sidak's multiple comparisons test (L) (variances were similar, and distribution was normal); ns, not significant; \**P* < 0.05; \*\**P* < 0.01; \*\*\**P* < 0.001.

Source data are available online for this figure.

A multidimensional scaling (MDS) plot showed that neonatal, naïve adult microglia, and microglia from EAE formed three separate and distinct global gene expression clusters, while CD11c-positive and negative subpopulations clustered relatively close together for each condition (Fig 3A). The major difference was therefore related to developmental age rather than subset phenotype.

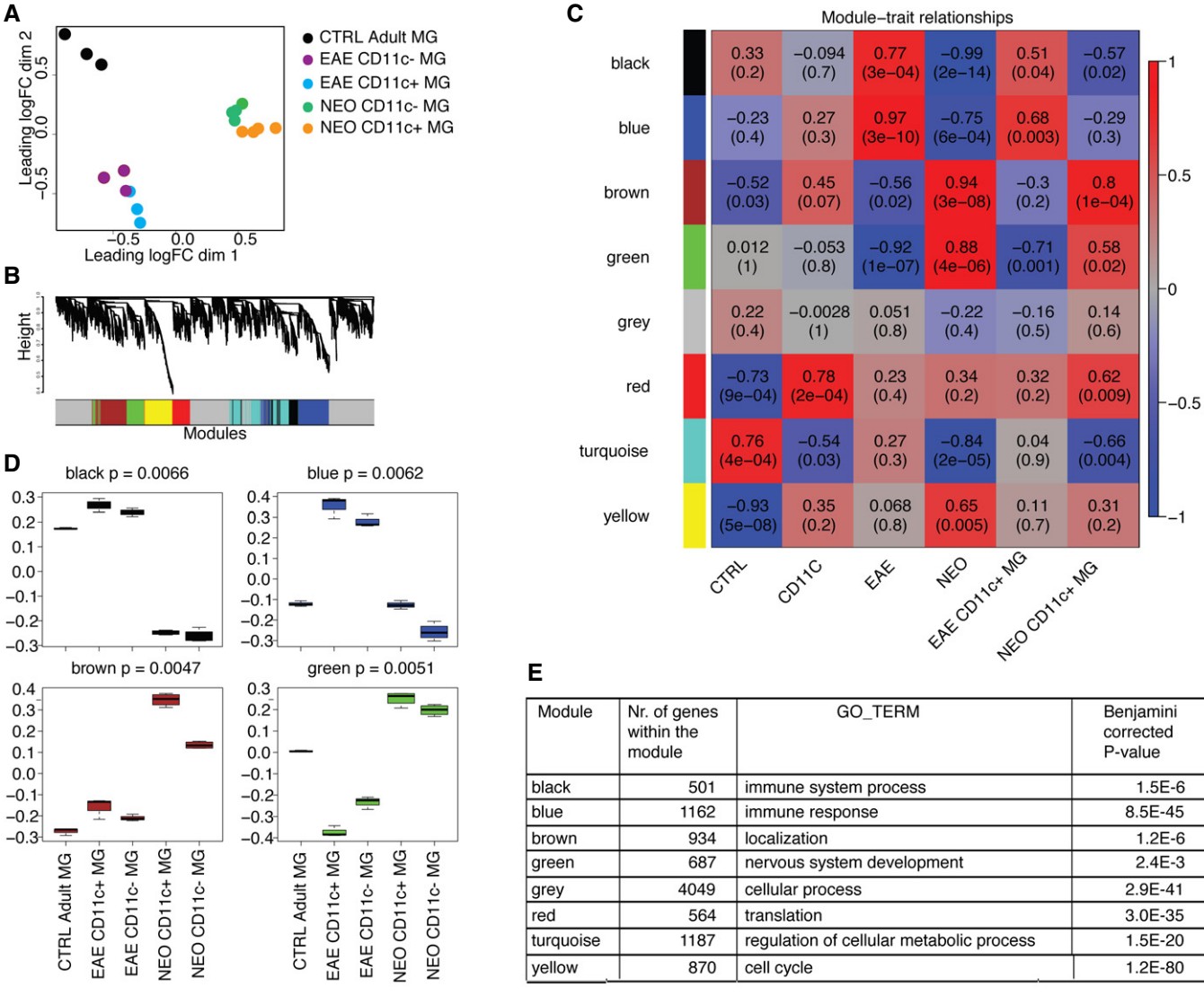

**Figure 3. Transcriptome analysis of neonatal, EAE, and adult microglia.**

A Multidimensional scaling shows that neonatal, EAE subpopulations of microglia and adult microglia have distinct transcriptional profiles. Colors indicate six different groups of samples: orange represents neonatal CD11c$^+$ microglia ($n = 4$), green neonatal CD11c$^-$ microglia ($n = 4$), blue EAE CD11c$^+$ microglia ($n = 3$), purple EAE CD11c$^-$ microglia ($n = 3$), and black adult microglia ($n = 3$). Each $n$ represents a pool of 10–15 mice from five individual EAE immunizations and four individual cell sorts of neonatal and naïve adult microglia.

B, C Co-expression networks were generated for 12,691 genes of the transcriptome dataset. Average linkage hierarchical clustering was applied to the topological overlap matrix and branches of highly correlating genes were formed, which were cut and assigned a color (B). For each module, the Module Eigengene (ME) was calculated, which represents the expression profile of the module. The ME values were correlated with binary variables (Spearman's correlation) that represent control, CD11c$^+$, EAE, neonatal, and CD11c$^+$ microglia in neonates and EAE. Within each table cell, upper values represent correlation coefficients between ME and the variable, while lower values in brackets correspond to Student asymptotic $P$-value (C).

D A boxplot containing the distribution of the black, blue, brown, and green ME values across the samples. The boxes contain the first and third quartiles; center line indicates the median and whiskers indicate minimum and maximum values. Kruskal–Wallis test was applied to determine whether ME values were significantly different between the groups.

E Table showing number of genes within the module and the top GO term for each module with Benjamini-corrected $P$-value.

In order to identify gene expression profiles associated with microglia from the different conditions, weighted gene co-expression network analysis (WGCNA) was applied to the RNA-seq data. In WGCNA, genes are clustered according to co-expression and a network with seven co-expression modules was identified (Fig 3B). From each module, the Module Eigengene (ME) was calculated, which is the first principal component and functions as a representative of the module. The ME values were correlated with variables that represent control adult microglia ("control"), CD11c$^+$ microglia population ("CD11c"), EAE microglia populations ("EAE"), neonatal microglia populations ("neonatal"), and CD11c$^+$ microglia from EAE ("EAE CD11c$^+$") and neonates ("neonatal CD11c$^+$") (Fig 3C) are depicted as a box plot per condition (Fig 3D).

With the exception of the gray module (that contained unclustered genes), all modules correlated significantly ($P < 0.005$) with some of the indicated variables. The ME of the yellow module was negatively correlated with "control" (−0.93; downregulated) and positively with "neonatal" (between 0.068 and 0.65). The top gene ontology (GO) enrichment category associated with the yellow module was the "cell cycle" term (Fig 3E). This suggested that microglia from neonates more abundantly expressed cell cycle genes, which is in line with the observed microglia proliferation during neurodevelopment (Matcovitch-Natan *et al*, 2016). In contrast, the turquoise module was negatively correlated with "neonatal" variable (−0.84) and positively correlated with "control" (0.76). This module was enriched for GO term "regulation of cellular metabolic process". The red module was not only negatively correlated with the "control", but also positively correlated with the "CD11c" variable (−0.73 versus 0.78). This suggested an activation-related increase in expression, which is more pronounced in CD11c-positive microglia. This red module was enriched for a "translation" GO term (Fig 3E), suggesting that neonatal and EAE microglia and in particular CD11c microglia were more translationally active.

Interestingly, four modules (black, blue, green, and brown) showed clear opposite correlations with the "EAE" and "neonatal" variables (Fig 3C). Where the blue and black modules increased their expression in EAE microglia, the expression of these modules was reduced in neonatal microglia (Fig 3D). The blue and black modules were enriched for "immune system process" and "immune response" categories. In contrast, ME values of the brown and green modules negatively correlated (Fig 3C) and were decreased (Fig 3D) in EAE microglia and positively correlated (Fig 3C) and increased (Fig 3D) in neonatal microglia. The green and brown modules were enriched for "nervous system development" and "localization" GO terms, respectively (Fig 3E). Additionally, the brown module also significantly correlated with the CD11c$^+$ neonatal microglia variable, suggesting that the CD11c$^+$ population in neonatal brains most abundantly expressed genes related to the "localization" GO term (Fig 3E). Taken together, these findings suggest that microglia under EAE conditions became immune-activated, while microglia in neonatal brains displayed a CNS-supportive and neurogenic phenotype.

## Distinct gene expression patterns in neonatal subsets of microglia and adult microglia

In order to further investigate differences between neonatal and adult microglia, the transcriptomes of CD11c$^+$ and CD11c$^-$ microglia from PN4-6 as well as of total adult microglia were compared. MDS plot showed that neonatal CD11c$^+$, neonatal CD11c$^-$, and adult microglia formed three separate and distinct global gene expression clusters (Fig 4A). From 12,691 genes expressed, 1,104 genes were increased and 1,001 decreased in neonatal CD11c$^-$ microglia compared to adult microglia. In neonatal CD11c$^+$ microglia, 1,633 genes were increased and 1,131 decreased, compared to adult microglia, and 641 were increased and 163 decreased, compared to CD11c$^-$ microglia (Fig 4B).

GO enrichment analysis on genes differentially expressed (logFC ≥ 1,5; FDR ≤ 0.01) in neonatal CD11c$^+$ and CD11c$^-$ microglia compared to adult microglia revealed significant enrichment ($P ≤ 0.01$ after Benjamini correction) for 55 GOTERM_BP_ALL

biological processes categories. Among those, some were associated with "cell cycle", "mitosis", and "cell division", suggesting that both neonatal microglia populations were in cell cycle or proliferating. Categories related to developmental processes including "nervous system development", "neurogenesis", "axonogenesis", and "neuron projection development" were significantly enriched in neonatal CD11c$^+$ microglia, but not in the CD11c$^-$ population (Fig 4C).

### Neonatal microglia express neuroectodermal genes

Neonatal microglia expressed neuroectodermal markers such as *Nes, Gfap, Pax6, Fabp5, Fabp7, Gap43, Vim, Dcx, Sox11,* and *NeuroD1* (Fig 5A). Nestin gene expression was compared in neonatal microglia, OPCs, astrocytes, and neurons from PN4-7 brains. All the cell populations expressed equal levels of nestin transcripts (Fig 5B). Nestin transcripts were also compared between neonatal CD11c$^-$ and CD11c$^+$ microglia, and adult microglia. While both neonatal microglia subsets expressed similar levels of nestin message, significantly less nestin transcripts were detected in adult microglia (Fig 5C). Consistent with these observations, flow cytometry analysis showed that around 30% of neonatal but only 3.7% of adult microglia expressed nestin (Fig 5D). Immunostaining revealed co-localization of nestin (Fig 5E) and GFAP (Fig 5F) with IBA1-positive cells in the neonatal brain.

### Neonatal CD11c$^+$ microglia are a major source of neurogenic signals

To further understand the difference between neonatal CD11c$^+$ and CD11c$^-$ microglia populations, GO enrichment analysis was performed on genes that were significantly more abundant (logFC ≥ 1,5; FDR ≤ 0.01) in neonatal CD11c$^+$ microglia compared to both neonatal CD11c$^-$ microglia and adult microglia. 251 genes fulfilled these criteria and showed significant enrichment ($P ≤ 0.01$ after Benjamini correction) for 14 GOTERM_BP_ALL biological processes categories. Interestingly, nearly 25% of the genes were annotated to the ontology term "developmental process" and "multicellular organismal development", 20% to anatomical "structure development" and "system development", 12% to "nervous system development", and < 10% to terms such as "cell motion", "cell migration", "cell adhesion", "response to external stimulus", and "response to wounding" (Fig 6A). Similar analysis for neonatal CD11c$^-$ microglia showed only 14 genes that fulfilled set criteria and the only significant enrichment was found for "immune effector process" category.

Importantly, oligodendrocyte supportive genes such as *Spp1* and *Igf1* were most abundant in CD11c$^+$ microglia. They significantly expressed *Bmp2,* a gene for a protein that promotes astrocyte differentiation, *Csf1* that induces microglial proliferation as well as genes involved in axonal guidance such as *Cxcl12, Ntn1, Sema7a,* and *Plxna2*. Moreover, CD11c$^+$ microglia expressed *Gpx3* and *Ptn* that promote neurite outgrowth, *Lgals1* promoting axonal growth, neuroprotective *Gpnmb, Adam19* known to shed neuregulin isoforms (Fig 6B), and tissue remodeling metalloproteinases: *Mmp12, Mmp19,* and *Mmp24* (not shown).

Gene expression levels of selected gene candidates encoding secreted proteins involved in neurodevelopment, including *Csf1, Spp1, Cxcl12, Gpx3, Ntn1,* and *Lgals1,* were compared in neonatal

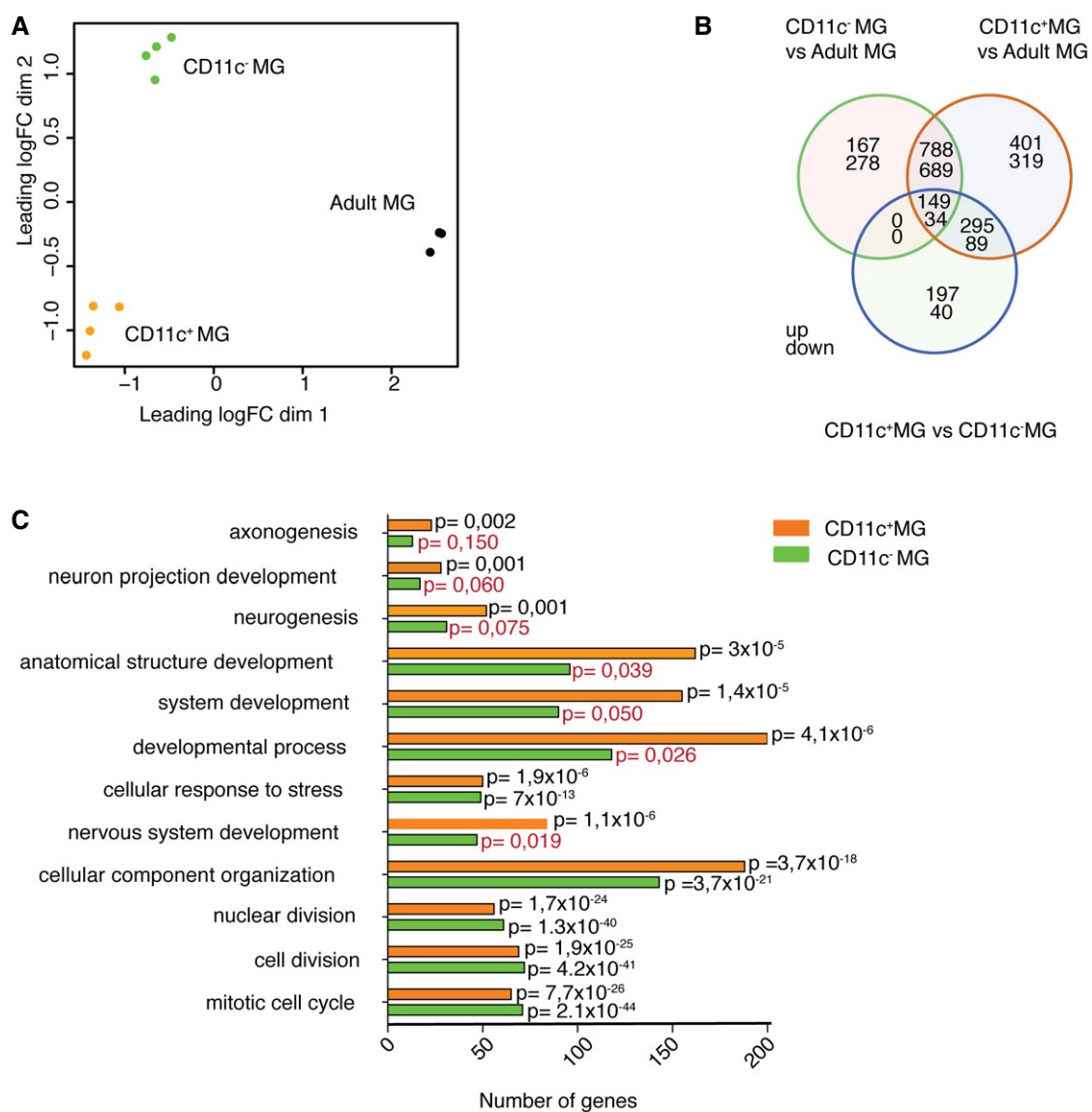

**Figure 4. Distinct gene expression profiles in neonatal and adult microglial subsets.**

A   Multidimensional scaling shows that neonatal subpopulations of microglia and adult microglia have distinct transcriptional profiles. Colors indicate three different groups of samples: orange represents neonatal CD11c⁺ microglia (*n* = 4), green neonatal CD11c⁻ microglia (*n* = 4), and black adult microglia (*n* = 3). Each *n* represents a pool of 10–15 mice from four individual cell sorts.

B   Venn diagram showing differentially expressed genes by neonatal CD11c⁺ microglia, neonatal CD11c⁻ microglia, and adult microglia. Numbers of genes differentially expressed comparing neonatal CD11c⁻ versus adult microglia, neonatal CD11c⁺ versus adult microglia, and neonatal CD11c⁺ versus neonatal CD11c⁻ microglia are indicated.

C   GO enrichment analysis of upregulated genes (logFC ≥ 1.5; FDR ≤ 0.01) in neonatal CD11c⁺ microglia (orange bars) and CD11c⁻ microglia (green bars) versus adult microglia identified significant (*P*-values ≤ 0.01 after Benjamini correction) enrichment for 55 GOTERM_BP_ALL biological processes categories, 12 of which are shown on the graph. Benjamini-corrected *P*-values are indicated on the bar graph; *P* ≥ 0.01 are marked with red font.

microglia, OPCs, astrocytes, and neurons, sorted from PN4-7 brains. These genes were expressed in all of the sorted cell populations. Microglia were the major source of *Spp1* and *Lgals1* (Fig 6C and D), and they also expressed significantly higher levels of *Gpx3* than astrocytes and neurons (Fig 6E). *Cxcl12* was equally expressed by all of investigated cell populations. *Ntn1* was expressed predominantly by OPCs and astrocytes and *Csf1* mainly by neurons, OPCs, and astrocytes (data not shown). *Spp1*, *Lgals1*, and *Gpx3* transcripts were further compared in CD11c⁺ and CD11c⁻ microglial

populations. CD11c⁺ microglia expressed much higher levels of *Spp1* (30-fold), *Lgals1* (sevenfold), and *Gpx3* (threefold) than their CD11c⁻ counterparts (Fig 6F–H).

## CD11c-targeted toxin regimens resulted in Itgax upregulation in neonatal brain

Having shown that CD11c⁺ microglia are a major source of many neurodevelopmentally important genes as well as being a critical

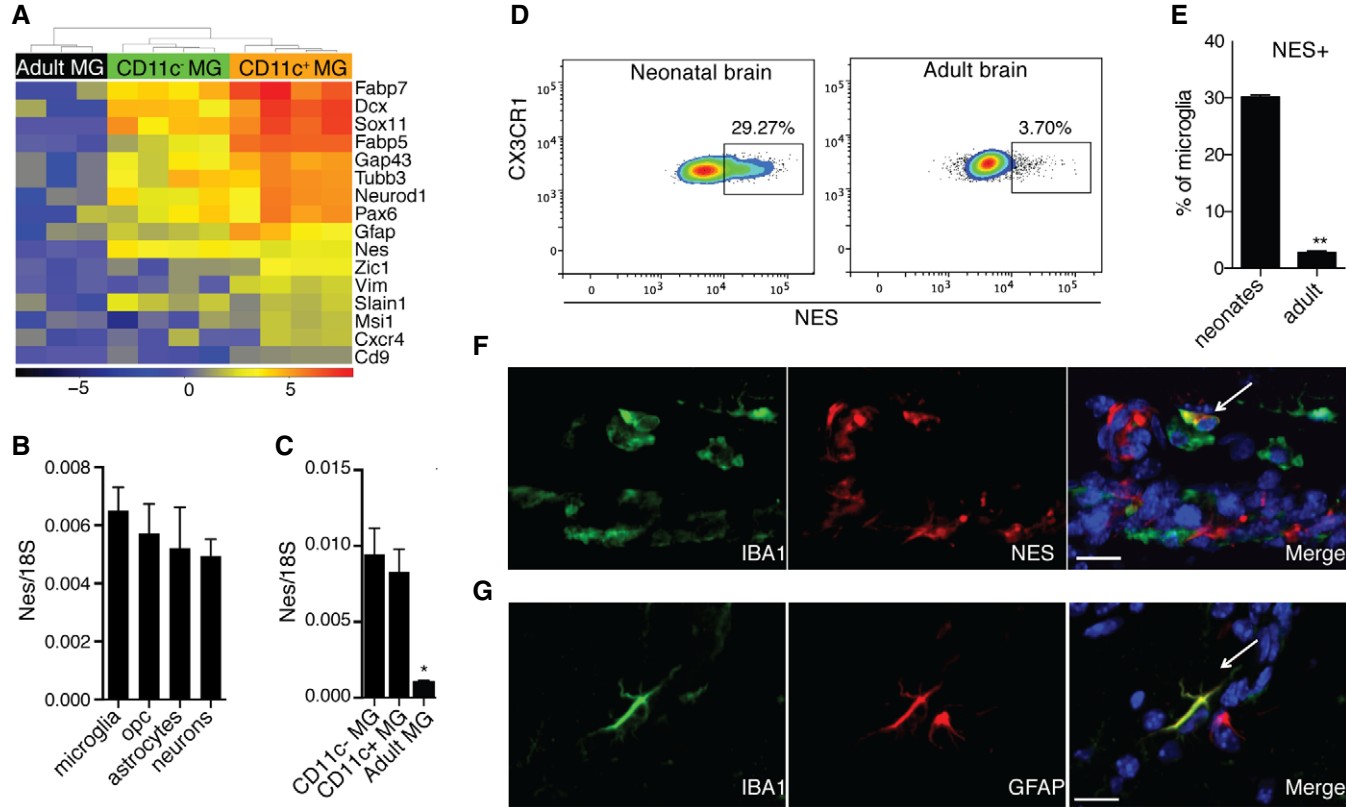

**Figure 5.   Neuroectodermal marker expression in neonatal microglia.**

A     A heatmap showing neuroectodermal gene expression in neonatal versus adult microglia. Scale represents log2 fold change normalized CPM expression values.

B, C   Expression of nestin in MACS-sorted microglia ($n = 4$), OPC ($n = 4$), astrocytes ($n = 4$), and neurons ($n = 3$) from brains of PN3-7 mice (B) as well as FACS-sorted CD11c$^+$ microglia ($n = 5$), CD11c$^-$ microglia ($n = 5$) from brains of PN4-7 mice, and total microglia from adult brain (C).

D, E   Representative flow cytometry profiles (D) and bar graph (E) showing nestin expression in microglia from brains of PN5 mice ($n = 5$) and adult mice ($n = 6$).

F, G   Confocal microscopic analysis showing co-localization of nestin (red) (F) or GFAP (red) (G) with IBA1 (green) ($n = 3$). Arrows point to IBA1, NES (F) and IBA1, GFAP (G) double positive cells. Scale bars = 15 μm.

Data information: Data are based on at least two experimental repeats. Data are presented as means ± SEM; each $n$ represents an individual mouse. $P$-values were determined by two-tailed Mann-Whitney $U$-test. ns, not significant; *$P < 0.05$; **$P < 0.01$.

source of IGF1 for myelinogenesis, we then asked the functional consequence of removal of these cells for developing brain. CD11c-iDTR mice that express diphtheria toxin receptor under control of a CD11c/*Itgax* promoter and WT littermates were injected with diphtheria toxin (DT) at days PN1 and PN3. Flow cytometry analysis, 48 h after the last DT injection, showed no difference in the percentage of CD11c$^+$ microglia compared to WT littermates (data not shown). Notably however, *Itgax* expression was significantly elevated in the brains of these mice, while *Igf1* expression did not change (Fig 7A). Similar findings were obtained in PN7 CD11c-DTA mice that express diphtheria toxin-A subunit under a CD11c/*Itgax* promoter—*Itgax* expression was significantly upregulated and *Igf1* expression was not altered (Fig 7B) in transgenic mice compared to WT littermates. In a separate approach, we injected saporin toxin conjugated to anti-CD11c antibodies into the corpus callosum and cerebellum of PN3 mice. Similar to the two other depletion strategies, we observed upregulation of *Itgax* and unchanged *Igf1* (Fig 7C) expression in comparison with isotype control-treated mice. Interestingly, we also observed a dramatic increase in CD11c$^+$ microglia

as a percentage of total microglia (Fig 7D and E) with unchanged absolute numbers (Fig 7F). A concomitant decrease in total microglia (Fig 7G) may be explained by CD11c$^-$ microglia having also expressed itgax and therefore becoming CD11c$^+$ in response to this stimulus. Taken together, the similarity in response of CD11c$^+$ microglia to three distinct toxin regimens underlines their unique status in the developing CNS.

### Clusters of repopulating microglia in adulthood contain CD11c$^+$ microglia

In contrast to developing CNS, where depletion of microglia has never exceeded 50% (Ueno *et al*, 2013), near-total ablation has been shown for adult mice (Bruttger *et al*, 2015). Recently, some of us demonstrated that 7 days after microglial ablation from adult CNS, using a CX3CR1$^{CreER}$ iDTR model, micro-clusters containing highly proliferating microglia were present throughout the CNS (Bruttger *et al*, 2015). We further analyzed these repopulating cells by RNA-seq and showed overexpression of *Itgax* (CD11c) and other genes

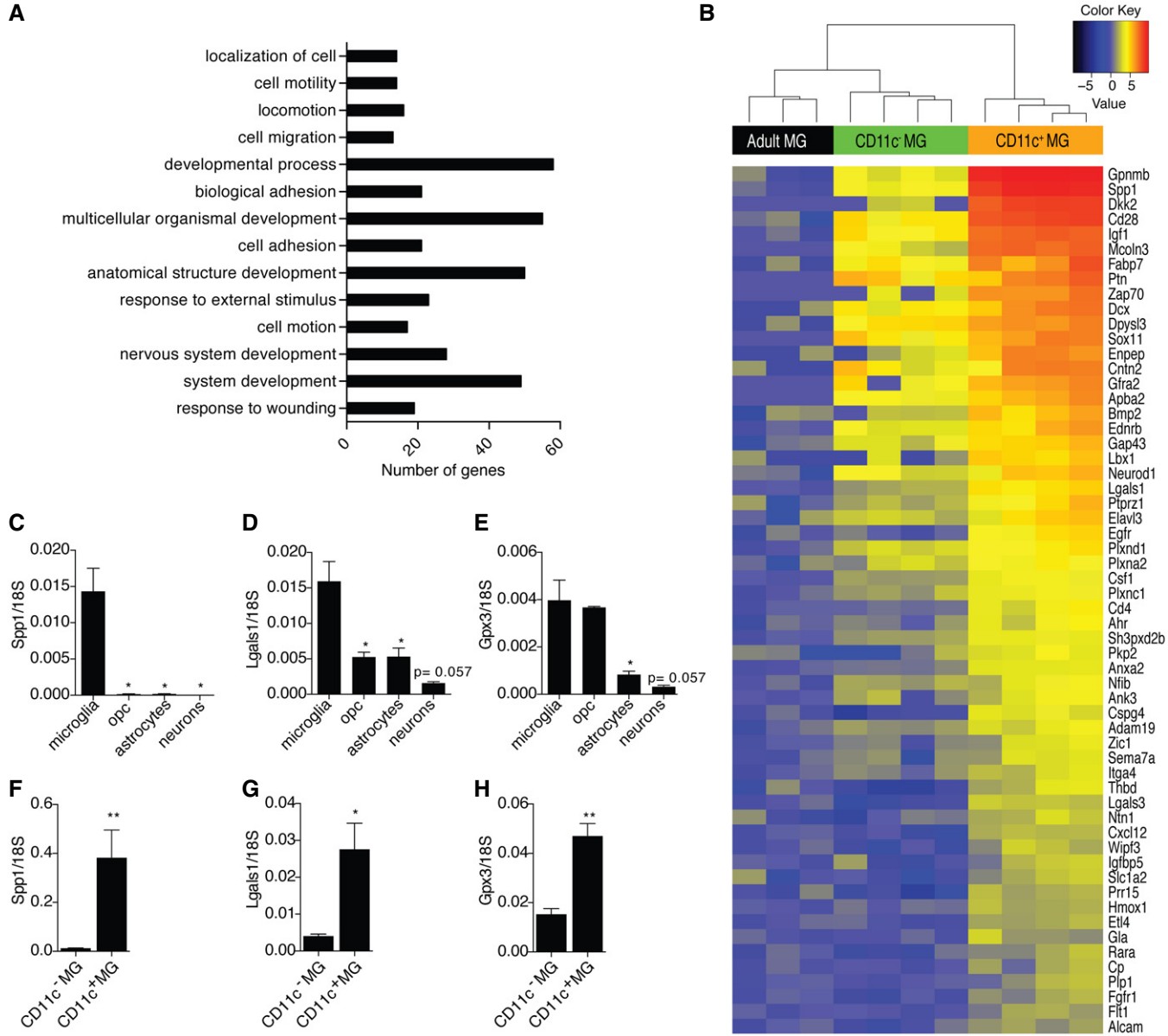

**Figure 6. Neonatal CD11c$^+$ microglia are a major source of neurogenic signals.**

A      Ontological analysis of upregulated genes (logFC ≥ 1.5; FDR ≤ 0.01) in neonatal CD11c$^+$ microglia in comparison with both neonatal CD11c$^-$ microglia and adult microglia identified significant ($P ≤ 0.01$ after Benjamini correction) enrichment for 14 GOTERM_BP_ALL biological processes categories.

B      A heatmap showing upregulated genes involved in system development in neonatal CD11c$^+$ microglia versus neonatal CD11c$^-$ and adult microglia. Scale represents log2 fold change normalized CPM expression values.

C–H   Expression of *Spp1*, *Lgals1*, and *Gpx3* relative to 18S rRNA in MACS-sorted microglia ($n = 4$), OPC ($n = 4$), astrocytes ($n = 4$), and neurons ($n = 3$) (C, D, E) as well as FACS-sorted CD11c$^+$ ($n = 5$) and CD11c$^-$ microglia ($n = 5$) (F, G, H) from brains of PN4-7 mice.

Data information: Data are based on at least two experimental repeats. Data are presented as means ± SEM; each $n$ represents an individual mouse. $P$-values were determined by two-tailed Mann–Whitney $U$-test. ns, not significant; $*P < 0.05$; $**P < 0.01$.

associated with CD11c$^+$ microglia such as *Igf1, Spp1*, and *Gpnmb*, in comparison with microglia from naïve adult mice (Fig 8A). Moreover, some clusters of repopulating microglia contained CD11c$^+$ cells (Fig 8B). These cells were also observed outside the clusters, likely migrating to and colonizing the tissue (Fig 8C). Interestingly, virtually all microglia in the clusters were nestin-positive (not shown). Next, we determined the overlap between the WGCNA

modules and up- and downregulated genes of the repopulating microglia. Genes that were upregulated in repopulating microglia, in comparison with control microglia, showed significant overlap for red ("translation") and blue modules ("immune system processes"), while downregulated genes for the blue module (Fig 8D). These data suggest that after ablation, repopulating adult microglia, similar to neonates, express nestin and CD11c but do not show a

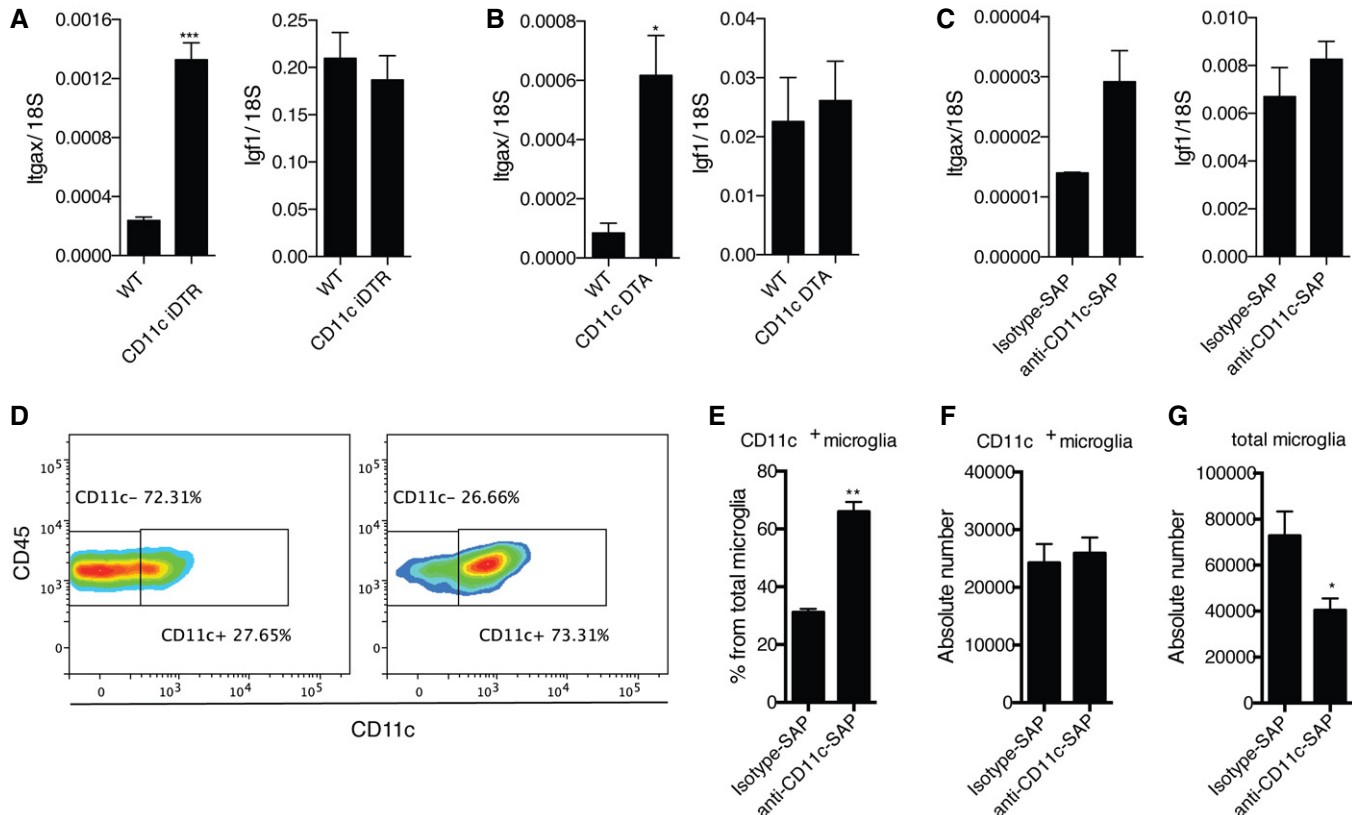

**Figure 7. Effect of CD11c-directed depletion strategies.**

A–C    Expression of *Itgax* and *Igf1* in PN5 brains of CD11c iDTR mice (*n* = 6) and WT littermates (*n* = 3) treated with DTx at days 1 and 3 after birth (A); PN7 brains of CD11c DTA mice (*n* = 5) and WT littermates (*n* = 3) (B); PN5 WT brains I.C. injected with saporin toxin conjugated with anti-CD11c (*n* = 6) or isotype control antibody (*n* = 5) (C).
D–G    Representative flow cytometry profiles (D) and flow cytometry analysis showing percentage from total microglia (E) and absolute numbers of CD11c⁺ microglia (F) as well as absolute numbers of total microglia (G) in PN5 WT brains I.C. injected with saporin toxin conjugated with anti-CD11c (*n* = 6) or isotype control antibody (*n* = 5).

Data information: Data are based on at least two experimental repeats. Data are presented as means ± SEM; each *n* represents an individual mouse. *P*-values were determined by two-tailed Mann-Whitney *U*-test. ns, not significant; *\*P* < 0.05; *\*\*P* < 0.01; *\*\*\*P* < 0.001.

neurogenic gene expression profile. Thus, the novel neuro- and myelinogenic CD11c⁺ microglial phenotype that we have described is unique to neonatal CNS.

## Discussion

We have identified a unique, myelinogenic phenotype of neonatal microglia. They are distinct from adult, EAE, and repopulating microglia. We have shown that a CD11c⁺ microglial subset significantly increases in early postnatal brain and then dramatically contracts as animals mature to adulthood. These microglia are a major cellular source of *Igf1*, *Spp1*, *Lgals1*, *Gpx3*, and other genes associated with neurogenic and myelinogenic processes in the neonatal brain, which is supported by effects of selective *Igf1* ablation from CD11c⁺ microglia on myelin gene expression and myelination in 3-week-old mice. These findings identify neonatal microglia, and especially the CD11c⁺ subset, as key players in neurodevelopment.

Microglia include functionally distinct subsets that can be distinguished by expression of the integrin CD11c, also known as complement receptor-4 (CR4). CD11c-expressing microglia have been found in several neuroinflammatory and neurodegenerative conditions possibly responding to CNS damage (Butovsky *et al*, 2006; Remington *et al*, 2007). We have previously shown that this subset of microglia expands in cuprizone-demyelinated corpus callosum (Remington *et al*, 2007; Wlodarczyk *et al*, 2015), in EAE (Wlodarczyk *et al*, 2014), and in neuromyelitis optica (NMO)-like pathology (Wlodarczyk *et al*, 2015). Now we show that the great majority of CD11c⁺ cells in the neonatal brain are microglia. Similar to microglia populations from EAE (Wlodarczyk *et al*, 2014, 2015), they express IBA1, CX3CR1 and, in contrast to blood-derived cells, do not express CCR2. Microglia have been recently shown to undergo temporal developmental stages characterized by different gene expression profiles (Matcovitch-Natan *et al*, 2016) and to mature in the second postnatal week (Bennett *et al*, 2016). Our data, in agreement with these studies, show that the gene expression profile of neonatal microglia significantly differed from adult homeostatic

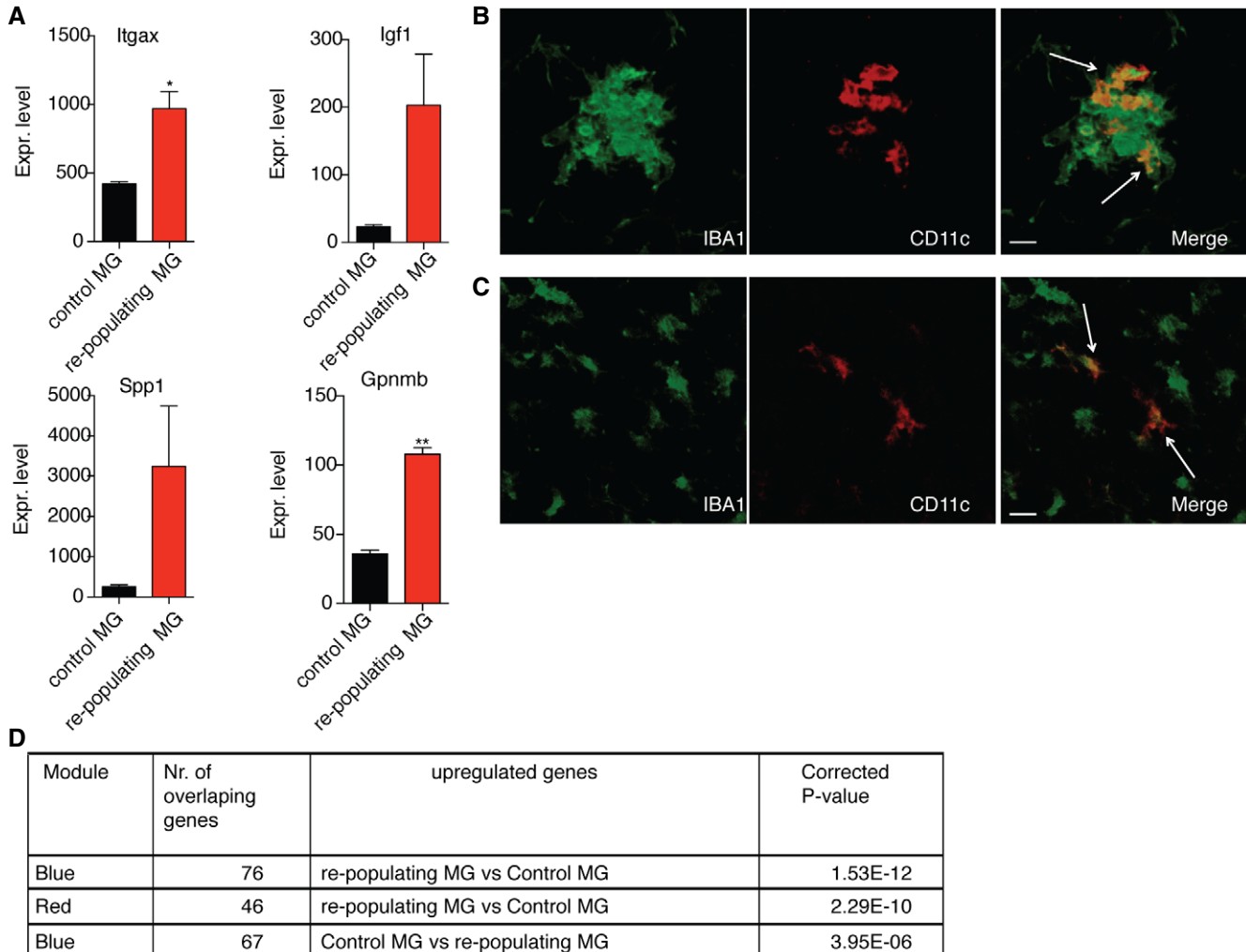

**Figure 8. Clusters of repopulating microglia in adult mice contain CD11c⁺ microglia.**

A    Gene expression values of *Itgax*, *Igf1*, *Spp1*, and *Gpnmb* in control microglia (n = 2) and repopulating microglia (n = 3) obtained from RNA-seq analysis. Each *n* represents a pool of two individual mice. *P*-values were determined by Student's *t*-test. ns, not significant; *$P < 0.05$; **$P < 0.01$.

B, C    Confocal microscopic analysis showing co-localization of CD11c (red) and IBA1 (green) within (B) and outside the repopulating microglia cluster (C) in brain stem of adult CX3CR1^CreER iDTR mouse 7 days after DTx treatment (n = 2). Arrows point to IBA1, CD11c double positive cells. Scale bars = 15 μm.

D    Table showing significant correlation of the up- and downregulated genes from repopulating microglia with two modules (red and blue) from Fig 3.

Data information: Data are based on at least two experimental repeats. Data are presented as means ± SEM.

microglia and in addition from that seen during EAE, showing myelin- and neurogenic in neonates versus an immune response signature in EAE. In line with the fact that neonatal mice have been shown to be resistant or to have delayed onset of EAE in comparison with adult mice (Smith *et al*, 1999; Cravens *et al*, 2013), our findings suggest that the neurosupportive capacity of microglia decreases with aging.

A role for microglia in neurodevelopment is increasingly accepted. Microglia have been reported to prune or sculpt developing synaptic connections, and a role for complement has been proposed (Stevens *et al*, 2007; Schafer *et al*, 2012). Microglia are defined by expression of CR3 (CD11b) and we show important anti-inflammatory and myelinogenic roles for CR4/CD11c-expressing microglia. We have not examined the role of complement, but

instead show dramatic and provocative correlation between expression of this receptor with myelinogenic capabilities, about which very little has been reported. We demonstrate that CD11c⁺ microglia expressed the majority of myelinogenic *Igf1* in the developing brain. IGF1 is necessary for neurodevelopment both in humans and in mice. Rare cases of IGF1 deficiency in humans are characterized by growth alteration, microcephaly, sensorineural deafness, and delayed psychomotor development (reviewed in Netchine *et al*, 2011). Deficiency of *Igf1* and its receptor in mice is usually postnatally lethal and those mice that survive to adulthood show microcephaly associated with increased neuronal death and myelination impairment (Beck *et al*, 1995). Expression of myelin-specific proteins as well as oligodendrocyte numbers in the developing corpus callosum is significantly reduced in *Igf1*-deficient mice (Ye

*et al*, 2002). In contrast, mice with transgenic overexpression of *Igf1* have larger brains, increased populations of neurons and oligodendrocytes as well as enhanced myelin production (reviewed in de la Monte & Wands, 2005). Neurons, OPCs, and microglia have all been reported to express *Igf1* (Liu *et al*, 1994; Mueller *et al*, 2013; Suh *et al*, 2013; Ueno *et al*, 2013) but their relative importance in postnatal development has not been defined so far. Microglial *Igf1* was recently shown to support layer V cortical neuron survival during postnatal development (Ueno *et al*, 2013). We have now shown that even partial depletion of *Igf1* in CD11c$^+$ microglia leads to reduction in brain weight, decrease in *Plp*, *Mbp*, *Mag*, and *Mog* expression in brain, and is associated with higher frequency of less myelinated fibers in corpus callosum. We showed that highly myelinating areas such as the developing corpus callosum and cerebellum are colonized specifically by CD11c$^+$ microglia, which further supports that through their high *Igf1* expression these cells influence myelin formation within these structures. Thus, as the major source of *Igf1*, the CD11c$^+$ subset of microglia plays a critical role in primary myelination and neuronal support in the neonatal CNS.

We show that neonatal microglia express many genes that are important for brain formation. Although genes associated with developmental processes were expressed in both subsets of neonatal microglia, this was much more pronounced in the CD11c$^+$ population. CD11c$^+$ microglia may contribute to shaping axonal organization in the CNS through expression of well-described axonal guidance signals such as *Cxcl12*, *Ntn1*, *Plxna2*, and *Sema7a*. They may stimulate neuronal and neurite outgrowth and astrocytogenesis through expression of *Ptn* (Garcia-Gutierrez *et al*, 2014), *Gpx3* (Buchser *et al*, 2012), and *Bmp2* (Nakashima *et al*, 1999), respectively, and affect remodeling processes in the developing brain tissue by expression of metalloproteinases *Mmp12*, *Mmp19*, and *Mmp24*. CD11c$^+$ microglia are therefore seen to be involved in promoting and directing organization of the developing CNS.

Microglia colonize the neuroepithelium very early during neurodevelopment (Ginhoux *et al*, 2010) and are autonomously maintained through proliferation (Ajami *et al*, 2007). Both CD11c$^+$ and CD11c$^-$ populations of microglia proliferated equivalently in response to neuroinflammation (Wlodarczyk *et al*, 2015). The mechanism by which the number of CD11c$^+$ microglia is regulated during brain maturation is obviously of key importance. Neonatal CD11c$^+$ microglia are themselves a source of *Csf1*, which suggests that these cells may represent a self-renewing population, since microglial development and maintenance is highly dependent on CSF1R signaling (Elmore *et al*, 2014), and mice deficient in CSF1R almost completely lack microglia (Ginhoux *et al*, 2010). Neonatal microglia upregulate genes involved in cell division and their number increases rapidly during the first postnatal days. Our findings show that CD11c$^+$ microglia survive and show a unique response to antibody-directed as well as cell-intrinsic and cell-extrinsic transgenic toxicity, and indirectly suggest that besides proliferation, they may also be reconstituted from the CD11c$^-$ microglia population.

Interestingly, both of the neonatal microglia populations that we studied, and therefore all neonatal microglia, express nestin at a similar level as other brain cell populations such as OPCs, astrocytes, and neurons. Nestin expression by neonatal microglia was transient and significantly decreased in adulthood. Microglia have been shown to express *Nes* and *Vim* also in naïve adult rat brain (Takamori *et al*, 2009), and nestin-expressing microglia have been shown to act as multipotential stem cells that give rise to neurons, astrocytes, or oligodendrocytes *in vitro* (Yokoyama *et al*, 2004). Recently, nestin expression has been demonstrated in repopulating adult microglia after their genetic ablation (Bruttger *et al*, 2015) and in inflammatory-activated microglia (Krishnasamy *et al*, 2017), pointing to nestin-positive microglia progenitors with high proliferative potential (Elmore *et al*, 2014). Here, we show that the clusters of such repopulating microglia are rich in CD11c$^+$ Nes$^+$ microglia. However, they do not show a neonatal-like gene signature, identifying the neonatal neurogenic phenotype of microglia as unique and transient.

The CD11c$^+$ microglial response is not restricted to neurodevelopment. Despite their immune response gene signature and being less myelinogenic/neurogenic than neonatal microglia, evidence suggests that they may be involved in regenerative processes in adulthood. CD11c$^+$ microglia express *Igf1* (Wlodarczyk *et al*, 2015) and *Lgals1* during EAE in adult animals, although this expression is significantly lower than in neonatal CD11c$^+$ microglia (Wlodarczyk, unpublished observations). Galectin-1 encoded by *Lgals1* promotes neurite outgrowth and has been reported to promote axonal regeneration (Horie *et al*, 1999; McGraw *et al*, 2004; Quinta *et al*, 2014). It also suppresses microglial activation (Starossom *et al*, 2012), promotes apoptosis of activated T cells, and has immunoregulatory properties in models of autoimmune diseases (Camby *et al*, 2006) including amelioration of EAE (Starossom *et al*, 2012). In the adult brain, only a small pool of CD11c$^+$ microglia remains, being available when needed for immunoregulation, regeneration, and remyelination, for instance in response to brain damage. Increase in this subset in EAE as well as in cuprizone-mediated acute demyelination and NMO-like pathology (Remington *et al*, 2007; Wlodarczyk *et al*, 2014, 2015) argues for stimulus-dependent response. Neonatal population dynamics were recapitulated in the adult brain, where the expanded population of cuprizone-induced CD11c$^+$ cells contracted dramatically upon withdrawal of the demyelinating stimulus (Remington *et al*, 2007). This suggests a more general response to loss of homeostasis, which will be further studied.

Taken together, we identify neonatal CD11c$^+$ microglia as a potent and inducible source of developmental proteins for neurogenesis, and myelinogenesis in the developing CNS.

## Materials and Methods

### Mice

C57BL/6j bom female mice aged 7–8 weeks were obtained from Taconic Europe A/S; CX3CR1$^{GFP/GFP}$ and CCR2$^{RFP/RFP}$, CD11c$^{Cre-GFP}$ (Stranges *et al*, 2007) and Igf1$^{fl/fl}$ were obtained from The Jackson Laboratory and maintained as a breeding colony (CX3CR1$^{GFP/GFP}$ and CCR2$^{RFP/RFP}$ were crossed with B6 mice to obtain heterozygotes) in the Biomedical Laboratory, University of Southern Denmark (Odense). All experiments performed in Denmark were approved by the Danish Animal Experiments Inspectorate (approval number 2014-15-0201-00369). CX3CR1$^{CreER}$, CD11c$^{Cre}$, DTA, and iDTR mice were housed in specific pathogen-free conditions and used in accordance with the guidelines of the Central Animal

Facility Institution (ZVTE, University of Mainz). Neonatal mice (p2-p21) used for experiments were of mixed sex.

## EAE model

Seven- to ten-week-old female mice were immunized by injecting subcutaneously 100 µl of an emulsion containing 300 µg of myelin oligodendrocyte glycoprotein (MOG)p35–55 (TAG Copenhagen A/S, Frederiksberg, Denmark) in incomplete Freund's adjuvant (DIFCO, Albertslund, Denmark) supplemented with 400 µg H37Ra *Mycobacterium tuberculosis* (DIFCO). *Bordetella pertussis* toxin (300 ng; Sigma Aldrich, Brøndby, Denmark) in 200 µl of PBS was injected intraperitoneally at day 0 and day 2. Animals were monitored daily from day 5 and scored on a 7-point scale as follows: 0, no symptoms; 1, partial loss of tail tonus; 2, complete loss of tail tonus; 3, difficulty walking; 4, paresis in both hind legs; 5, paralysis in both hind legs; and 6, front limb weakness. Due to ethical consideration, mice were sacrificed when they reached grade 6 or 24 h after hind legs paralysis.

## Microglia depletion strategies

For CD11c$^+$ microglia depletion, CD11c-iDTR mice and WT littermates were injected subcutaneously with 25 ng/g body weight of DT 1 and 3 days after birth and were sacrificed 5 days after birth.

C57BL/6j bom PN3 pups were randomized into two groups and injected intracerebrally into two areas of the brain (coordinates: 2.2 mm anterior, 1.0 mm ventral and 1.0 posterior, 1.2 ventral) with 1.3 µg of biotinylated anti-CD11c antibody (N418; eBioscience) or biotinylated anti-hamster IgG (Jackson ImmunoResearch) conjugated to streptavidin-saporin (ATS) prepared according to the manufacturer's protocol. Pups were sacrificed 2 days after injection.

For microglia depletion in CX3CR1$^{CreER}$ iDTR mice, first 2 mg tamoxifen was administered subcutaneously (s.c.) twice on postnatal days 12 and 14, and then at age of 8 weeks, mice were injected intraperitoneally (i.p.) with 500 ng diphtheria toxin (DT; Merck Millipore) three times, with a 1-day interval between each injection (Bruttger *et al*, 2015).

## Magnetic-activated cell sorting (MACS)

To isolate cells from brain PN3-7, mice were sacrificed, brain tissue was collected, and single cell suspensions were generated using Neural Tissue Dissociation Kit (P) (Miltenyi Biotec) for OPC, CD11b$^+$ cells (microglia), and astrocyte isolation, or Neural Tissue Dissociation Kit -Postnatal Neurons (Miltenyi Biotec) for neuron isolation. Cells were isolated by magnetic separation using CD140a (PDGFRα) MicroBead Kit (Miltenyi Biotec), CD11b (Microglia) MicroBeads (Miltenyi Biotec), anti-ACSA-2 MicroBead Kit (Miltenyi Biotec), or Neuron Isolation Kit (Miltenyi Biotec). All steps were done according to the manufacturer's protocols.

## Fluorescence-activated cell sorting (FACS) and flow cytometry

To isolate microglia from brain, PN2-PN5, PN7, PN28, and 8- to 12-week-old mice were anaesthetized with 200 mg/kg of pentobarbital and intracardially perfused with ice-cold PBS. Brain tissue was collected and a single cell suspension was generated by forcing through a 70-mm cell strainer (BD Biosciences). Mononuclear cells

were collected after centrifugation on 37% Percoll (GE Healthcare Biosciences AB). They were first incubated with anti-Fc receptor (Clone 2.4G2; 1 mg/ml; BD Pharmingen) and Syrian hamster IgG (50 mg/ml; Jackson Immuno Research Laboratories Inc.) in PBS, 2% fetal bovine serum (FBS), then with anti-CD45 (Clone 30-F11; Biolegend), anti-CD11b (Clone M1/70; Biolegend), and biotin-conjugated anti-CD11c (Clone HL3; BD Pharmingen) antibodies in PBS, 1% FBS, and finally with streptavidin-APC (Biolegend). Cell populations were gated based on isotype-matched control antibodies as CD45$^{dim}$ CD11b$^+$ CD11c$^-$ (CD11c$^-$ microglia), CD45$^{dim}$ CD11b$^+$ CD11c$^+$ (CD11c$^+$ microglia) and sorted on a FACSAria™ III cell sorter (BD Biosciences) or data were collected on an LSRII™ flow cytometer (BD Biosciences) and analyzed using FACSDiva™ software version 6.1.2 (BD Biosciences). For the analysis of nestin expression after extracellular staining (as above), cells were fixed and permeabilized using BD Cytofix/Cytoperm™ according to the manufacturer's protocol and stained with anti-nestin and corresponding isotype control (Clone 307501; R&D Systems).

## RNA extraction, quantitative RT–PCR (RT–qPCR)

FACS-sorted CD11c$^+$ and CD11c$^-$ microglia were placed in RLT buffer (Qiagen) and total RNA was extracted using RNeasy columns as per the manufacturer's protocol (Qiagen). MACS-sorted neurons, OPCs, CD11b$^+$ cells, and astrocytes were placed in QIAzol Lysis Reagent and total RNA was extracted using miRNeasy Micro Kit according to the manufacturer's protocol (Qiagen). Reverse transcription was performed with M-MLV reverse transcriptase (Invitrogen) according to the manufacturer's protocol. Quantitative real-time PCR (qPCR) was performed with 1 µl cDNA in a 25 µl reaction volume containing Maxima$^®$ Probe/ROX qPCR Master mix (Fermentas), TaqMan$^®$ PreAmp Master Mix Kits for *Cxcl12*: Mm00445553_m1, *Ntn1*: Mm00500896_m1, *Lgals1*: Mm00839408_g1, *Gpx3*: Mm00492427_m and *Nes*: Mm00450205_m1, *Spp1*: Mm00436767_m1, *Mog*: Mm00447824_m1, *Plp*: Mm01297210_m1, *Mbp*: Mm01266402_m1, *Mag*: Mm00487538_m1. For *Igf1*, Maxima$^®$ SYBR Green/ROX qPCR Master Mix (2X) Probe/ROX qPCR Master Mix (Fermentas) with forward and reverse primers (800 nM; from ATGC) were used. *Igf1* primer sequences were as follows: For: CCG AGG GGC TTT TAC TTC AAC AA; Rev: CGG AAG CAA CAC TCA TCC ACA A. PCR were done on an ABI Prism 7300 Sequence Detection System (Applied Biosystems). Results were expressed relative to *18S* rRNA ($2^{ΔCT}$ method) as endogenous control (TaqMan$^®$ Ribosomal RNA control Reagents kit; Applied Biosystems). cDNA was diluted 1/500 for *18S* rRNA analysis.

## Genomic PCR

DNA was extracted by DNeasy Blood & Tissue Kit (Qiagen) according to the manufacturer's protocol, and quantitated using NanoDrop (ThermoFisher). To detect Cre-induced recombination, PCR on 90 ng DNA was performed as described previously (Liu *et al*, 1998) using ID-3: CACTAAGGAGTCTGTATTTGGACC; ES-1: AGCCTCTC AACTAAGACAATA primers.

## RNA sequencing

RNA quality check was done using an Agilent bioanalyzer, and only high-quality samples were used (RIN > 7) for sequencing. Sequence

libraries were prepared using the Illumina Truseq RNA sample preparation kit. RNA sequencing was performed at the Genome Analysis Facility of the University Medical Center Groningen with an Illumina Hiseq 2500. Single-read 50-bp sequencing was done, aligned to the ensemble reference genome using the Star 2.3.1 l aligner (Dobin *et al*, 2013), allowing for two mismatches. Samtools 0.1.19 was used to sort the aligned reads (Roussos *et al*, 2012). Quantification was done using HT-seq count 0.5.4 (Anders *et al*, 2015). Between 6 and 11 million uniquely aligned reads per sample were generated. Count data were loaded in R and analyzed with the EdgeR BioConductor Package (Robinson *et al*, 2010). Differential gene expression was done between groups of interest (Dataset EV1). Heatmaps were generated with heatmap2 function of package gplots. Functional annotation was done using DAVID (Huang *et al*, 2009).

Detailed procedures for RNA-seq of repopulating microglia including data analysis are described in Bruttger *et al* (2015).

## Weighted gene co-expression network analysis (WGCNA)

WGCNA (Langfelder & Horvath, 2008) was applied to the count per million (CPM) expression data. A co-expression network was constructed with a beta value of 20. Genes were clustered into branches of highly expressed genes, and modules were identified with the tree cut algorithm with the additional PAM stage. Six binary variables were generated that were used to calculate the module trait relationships in which all groups were set to zero with the exception of particular groups of interest: control (1's for microglia obtained from healthy control brain), CD11c (1's for both EAE CD11c and neonatal CD11c), EAE (1's for CD11c negative and microglia obtained from EAE brains), neonatal (1's for CD11c negative and microglia obtained from neonatal brains), CD11c EAE, and CD11c neonatal. In addition, a between-group Kruskal–Wallis test was applied to the Module Eigengene values, to detect overall differential expression. Modules were functionally annotated using DAVID (Huang *et al*, 2009).

## Histology

Seventy-micrometer or twelve-micrometer sections from 4% PFA-fixed, frozen brains, or spinal cords of perfused mice were cut on a cryostat and stored in de Olmos cryoprotectant solution (Khorooshi & Owens, 2013) at −14°C or in −20°C on Superfrost Plus slides (Thermo Scientific), respectively. Sections were washed in PBS and incubated for 10 min in ice-cold acetone or 30 min with 10% methanol, 10% $H_2O_2$ in PBS to block endogenous peroxidase. After repeated rinses with 0.2% Triton X-100 in PBS (PBST), they were incubated for 1 h in 3% BSA in PBS to block unspecific binding. Next, sections were incubated overnight at 4°C with corresponding primary antibodies: rabbit anti-IBA1 (Cat. No. 019-19741 Wako), rat anti-nestin antibody (7A3; Abcam), hamster anti-CD11c (N418; Bio-Rad), anti-myelin PLP (Cat. No ab105784; Abcam), and rabbit anti-laminin (Cat. No. CL54851AP; Cedarlane). Following primary antibody incubation, the sections were washed with PBST and incubated for 1 h with the appropriate secondary antibody: goat anti-Armenian hamster IgG-HRP (Cat. No sc-2904; Santa Cruz Biotechnology), donkey anti-rabbit 488 (Cat. No A21206; Invitrogen), donkey anti-rat Alexa 594 (Cat. No A21209; Invitrogen), and goat

anti-rabbit Alexa 647 (Cat. No A21245; Invitrogen). Following secondary antibody incubation, the sections were washed with PBS and incubated for 4 min with TSA™ Plus Cy3 System (PerkinElmer).

After immunofluorescent labeling, nuclei were visualized by DAPI staining and the sections were mounted with Fluorescence Mounting Medium (DAKO). Visualization of GFP was done as previously described (Khorooshi *et al*, 2015). The sections were visualized on an Olympus FV1000MPE confocal microscope and analyzed by FV10-ASW 4.06 software.

## Electron microscopy

Animals were sacrificed and transcardially perfused with PBS followed by a fixative containing 4% paraformaldehyde and 1.5% glutaraldehyde. After postfixation of the brains in the same fixative, sagittal sections were acquired using a vibrating microtome (Leica Microsystems, Wetzlar, Germany) with a thickness of 75 μm. The sections were stained with 0.5% osmium tetroxide in PBS for 30 min and subsequently rinsed in PBS followed by dehydration with 30, 50 and 70% ethanol. Afterward, the sections were treated with 1% uranyl acetate in 70% ethanol for 1 h, before the tissue was further dehydrated by using 80, 90, 96, 100% ethanol and finally propylene oxide. The sections were transferred in Durcupan (Sigma Aldrich, Steinheim, Germany) and embedded in between coated microscope slides and cover slips before the polymerization process at 56°C for 48 h. Regions of interest were located by light microscopy, marked, and transferred on blocks of resin for a second polymerization step. Finally, the embedded tissue was cut into semi-thin sections and areas for ultrastructural analysis were cut into 55-nm ultra-thin sections using an ultra-microtome (Leica Microsystems, Wetzlar, Germany) and transferred on Formvar-coated grids and stained with lead citrate for 6 min. The analysis was performed using Zeiss SIGMA electron microscope (Zeiss NTS, Oberkochen, Germany).

Four different regions of the corpus callosum from each sample were analyzed. G-ratios of transversally sectioned axons were calculated using ImageJ software and compared between groups. The analysis was performed by an investigator who was blinded to the experiment.

## Statistical analysis

No statistical methods were used to predetermine sample sizes, and exact group numbers were determined by animal availability. We ensured that the sample sizes were similar to those generally employed in the field. For each EAE experiment, at least 16 mice were immunized (av. 50% of mice show EAE symptoms required to enter the experiment). In order to collect sufficient amount of RNA for RNA-seq analysis, each sample consisted of 10–15 pooled mice.

To obtain unbiased data, analysis of the samples was performed blinded to the genotype. Only after finalization of all quantitative measurements, the samples genotypes were decoded.

All experiments were repeated at least twice, and data are presented as means ± SEM. Statistical significance was assessed using the two-tailed Mann–Whitney *U*-test (GraphPad Prism 6) unless specified otherwise. Data met assumptions for the used tests.

Normal distribution was assessed by Shapiro–Wilk normality test. *P*-values < 0.05 were considered significant.

### Data availability

The datasets of RNA-seq from adult, EAE, and neonatal microglia are deposited in Gene Expression Omnibus (GEO): GSE78809, and will become available via the Glia Open Access Database (Holtman *et al*, 2015): www.goad.education. The datasets of RNA-seq of repopulating microglia are described in Bruttger *et al* (2015) and deposited in GEO: GSE68376.

**Expanded View** for this article is available online.

### Acknowledgements

This work was supported by grants from Lundbeckfonden, Danish Council for Independent Research, Danish MS Society, and Warwara Larsens Fond. We thank Dina Arengoth, Pia Nyborg Nielsen, Kirstine Nolling Jensen, and Nieske Brouwer for expert technical assistance and Inger Andersen and Lars Vitved for help with cell sorting. The bioimaging experiments reported in this paper were performed at DaMBIC, a bioimaging research core facility, at the University of Southern Denmark. DaMBIC was established by an equipment grant from the Danish Agency for Science Technology and Innovation and by internal funding from the University of Southern Denmark.

### Author contributions

AWl, TO, AWa, and BJLE designed the work. AWl, AB-B, IK, RK, NY, NAM, JJB-B, JB, KK, and MK performed the experiments. AWl, IRH, MK, EWGMB, BJLE, and JB analyzed the data. AWl and TO wrote the manuscript. All authors read and approved the final manuscript.

### Conflict of interest

The authors declare that they have no conflict of interest.

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
