## [Review Process File · The EMBO Journal]

Manuscript EMBO-2016-96056

A novel microglial subset plays a key role in myelinogenesis in developing brain

Agnieszka Włodarczyk, Inge R Holtman, Martin Krueger, Nir Yogev, Julia Bruttger, Reza Khorooshi, Anouk Benmamar-Badel, Jelkje J. de Boer-Bergsma, Nellie A Martin, Khalad Karram, Isabella Kramer, Erik W.G.M. Boddeke, Ari Waisman, Bart J.L. Eggen, Trevor Owens

Corresponding author: Trevor Owens, University of Southern Denmark

Review timeline:

Submission date:	10 November 2016
Editorial Decision:	05 December 2016
Revision received:	04 May 2017
Editorial Decision:	27 June 2017
Revision received:	13 July 2017
Editorial Decision:	28 July 2017
Revision received:	29 August 2017
Editorial Decision:	30 August 2017
Revision received:	30 August 2017
Accepted:	31 August 2017

Editor: Karin Dumstrei

Transaction Report:

1st Editorial Decision

05 December 2016

Thank you for submitting your manuscript to The EMBO Journal. Your study has now been seen by three referees and their comments are provided below.

As you can see from the comments, referee #1 and 2 find the analysis insightful and interesting while referee #3 is more critical. All three referees raise a number of specific concerns that are relevant and valid. The points are clearly outlined below.

Given the comments provided, I would like to invite you to submit a suitably revised manuscript for our consideration. I should add that it is EMBO Journal policy to allow only a single round of revision, and that it is therefore important to address the raised issues at this stage. Let me know if we need to discuss any of the points further- happy to do so.

When preparing your letter of response to the referees' comments, please bear in mind that this will form part of the Review Process File, and will therefore be available online to the community. For more details on our Transparent Editorial Process, please visit our website: http://emboj.emboipress.org/about#Transparent_Process

We generally allow three months as standard revision time. As a matter of policy, competing

manuscripts published during this period will not negatively impact on our assessment of the conceptual advance presented by your study. However, we request that you contact the editor as soon as possible upon publication of any related work, to discuss how to proceed. Should you foresee a problem in meeting this three-month deadline, please let us know in advance and we may be able to grant an extension.

Thank you for the opportunity to consider your work for publication. I look forward to your revision.

REFEREE REPORTS

Referee #1:

The study by Wlodarczyk and colleagues describes 1) a new microglia subpopulation that is developmentally regulated, 2) a novel function of microglia in general that is beyond the already described text book knowledge on microglia during brain development that consists of their function for neurogenesis, synaptic pruning, neuronal nutrition and others. Thus, the topic tackled by the authors is of great importance and some of their findings are of big importance.

Despite the relevance of the topic there are still some weaknesses of the study. Some of their statements are not justified and simply not supported by the data. Further, the flow of the manuscripts should also be improved.

Major concerns:

1. The main message of the paper that CD11c+ microglia are important for myelinogenesis needs to be supported by further data. I strongly suggest to perform myelin thickness measurements (G-ratio) on myelinating axons by electron microscopy in CD11cCre IGF1fl/fl animals. This important experiment will further strengthen their main message.
2. Figure 1: The authors state that their CD11b+CD11c+CD45high cells are microglia that is very bold based only on the FACS data. In fact, the expression of CCR2 in this cell population is untypical for microglia and usually restricted to circulating cells. This surface marker profile could also indicate other cells such as monocytes, dendritic cells and activated CNS macrophages. This, the authors should be more careful on their wording until this point. They provide confirmation for the microglia nature of these cells later in Fig. 3 that should therefore be moved already to figure 1.
3. The authors state several times in the text (eg. abstract, title page 9 etc.) that CD11c+ microglia have "stem cell-like potential". However, there is simply no experimental evidence for this strong statement because only transcriptional data are shown.
4. Similarly, the role of this microglia subset was described as "key role in...neurogenesis" (see title) but no evidence is provided for this.
5. Their statement that CD11c+ microglia "are self-renewing" (page 11) is not supported by any data and should be omitted.
6. It is illogical and not clear why CD11c+ microglia can not be depleted by their DTR approach in CD11cCre DTR mice. The lack of depletion points to technical difficulties.

Minor points:

1. First sentence in the introduction. Please add references for microglia origin, namely Ginhoux et al. Science (2010, already in the reference list) Schulz C et al. Science (2012), Kierdorf et al. Nat Neurosci (2013). The last two references were missing at all.
2. Figure 2: Labelling of the pictures is incorrect. "Tg" should be changed to the correct genotype CD11cCre IGF1fl/fl.

Referee #2:

In this well-written manuscript by Wlodarczyk and colleagues, the authors study the role of CD11c-expressing microglia in myelination and provide transcriptomic analyses of this potentially distinct microglia subset. They nicely demonstrate that more microglia are CD11c+ during development and, based on previous work, after EAE. They conclude that loss of IGF1 in CD11c+ microglia

causes a decrease in myelin protein expression and that there are several cassettes of genes that are expressed more highly by CD11c+ but not CD11c- microglia. If successful, this work is of great potential interest, given the recent implications of microglia in synapse elimination, neuronal survival, and roles in human disease. Indeed, the idea of microglia heterogeneity is a tantalizing one. There remain several major concerns, however, that if addressed would greatly increase the impact of their findings and better validate their conclusions.

1) CD11c+ cells as a distinct population of microglia. The authors nicely demonstrate that there are pockets of CD11c+ cells throughout the developing, and in the injured CNS. It remains unclear to this reviewer: are all microglia somewhat CD11c+, with some small variability in expression? The FACS plots of developing microglia don't show as nice of a clear CD11c+ population as this lab has shown in the group's previous work in EAE. The selected immunostaining images give the impression (I'm guessing falsely) that all microglia are CD11c+. Some additional images could be helpful to clarify. Regarding the issue of if these cells are distinct: the sequencing and qPCR data throughout and specifically for the CD11c+ versus CD11c- microglia require some orthogonal methods for validation, such as proof of good isolation (microglia-specific versus glia or neuronal gene expression) and RNA in situ validation of some differences (in particular true GFAP expression in CD11c+ microglia). It was difficult from the FC data provided what the absolute expression values were for sequencing - especially since other astrocyte-, oligodendrocyte-, and neuron-specific genes were also upregulated in CD11c+ cells according to the provided Supplemental Table 1, implying the possibility of differential cell-type contamination instead of the differential expression of non-microglial genes such as *Fabp7*, *Neurod1*, and *GFAP*.

2) CD11c+ microglia-derived IGF1 responsible for myelination. PLP immunostaining, while one marker of myelin protein expression, is not necessarily a great way to show decreased myelination. Further, there is no mechanistic insight - either through survey of the literature about IGF1's role in myelination nor about how CD11c+ microglia versus others might play this role. A) Is the decrease in PLP and other myelin protein sufficient to demonstrate a defect in myelination? How does it compare with the IGF1 full null mouse? Or in a mouse with IGF1 loss in all microglia? B) Is the effect via survival of the oligodendrocytes, precursors, or perhaps the neuronal survival effect on projection neurons? Either the author's conclusions should be stepped back or they should provide some mechanistic insight. At least, the authors should show the A) effect on myelination either through another method (EM, physiology, et)? B) # of CD11c-Cre-GFP+ cells in the brain (with staining to show this Cre targets only a subset of microglia and C) Verify the loss of IGF1 only in CD11c+ cells. They show global levels, but other drivers, such as *CX3CR1-Cre* have been shown to have some off-target recombination in neurons developmentally, and given that the authors claim these CD11c+ microglia as a "novel" population, more description of this mouse is helpful and warranted. This could be accomplished by crossing in a recombination reporter. These would also be helpful for the depletion studies, as it would help distinguish robust escaper cells from cells that were impervious to DTA depletion.

In addition to these major points, there are several smaller points:

- 1) Please include n used; on occasion a representative image of an experiment is shown from 3 individual experiments, but not reported as such. It is not always clear what counts as N either, especially for sequencing.
- 2) Absolute counts would be helpful addition to supplemental table 1.
- 3) Are CD11c+ cells more proliferative after injury or depletion? Given their robust appearance after, it could be that proliferating cells are more likely CD11c high?

Referee #3:

This is a rather descriptive study with no clear convincing data as well as clear overstatements/over-interpretation of the data presented. The conclusion that the CD11c+ population is a myelinogenic and neurogenic phenotype with stem cell-like potential is clearly an overstatement. The data presented here is just correlative and based mostly on gene expression profile. In addition, the conclusion that "Inability to deplete CD11c+ microglia from the developing brain further underlines their importance in neurodevelopment" is not accurate. The authors did not well characterize the depletion specificity and efficiency of their model. The only conclusion possible from their data is that there are not able experimentally to deplete neonatal CD11c+ microglia.

Below are some comments.

The flow cytometry data are not presented at the highest publication standards. The authors need to present the flow plots in biexponential display as well as provide a better CD11c staining with appropriate control. In this regard, the CD11c staining is not resolutive! The authors need to show an isotype control to determine where the CD11c population is really starting....as the staining looks rather as a continuum than a define population.

In the Figure 1 immunohistostainings, the CD11c+ population appears not as homogeneously distributed as the usual microglia population. Can the authors document this better? Could the CD11c+ population be in fact perivascular macrophages? Interestingly, the cells are not ramified at all.

Figure 2: mislabeling of the figure legend. There is no k, l and m panels.

Selective ablation of IGF1 under CD11c+ cells is not convincing. The conclusion "selective depletion from CD11c+ microglia leads to impairment of primary myelination. " is not correct as the authors did not formally show that IGF was selectively ablated only in CD11c microglia in these experiments (what about other CD11c+ cells?). They need to sort the cells and track carefully recombination. The graph of the quantification of PLP staining intensity is also misleading as the Y axis starts at 70, suggesting a minimal effect.

The Nestin observation is interesting but is not related to the CD11c+ vs CD11c- microglia study as both population express the same level of Nestin. Same for the repopulating experiments, the authors show that some repopulating microglia are CD11c+ but do not provide any quantification.

Again concluding that "Taken together, this inability to deplete CD11c+ microglia using three distinct approaches underlines their critical importance for development of the CNS" is not possible. There is simply no depletion.

Anout the fact that some clusters of repopulating microglia contained CD11c+ cells, until the authors show that these cells are related to the CD11c+ cells that they are studying in development, nothing can be concluded. They could be totally unrelated cells that just upregulate CD11c expression due environmental cues.

About a role of microglia during development, the authors should cite Squarzzoni et al, Cell Reports.

1st Revision - authors' response

04 May 2017

Referee #1:

The study by Włodarczyk and colleagues describes 1) a new microglia subpopulation that is developmentally regulated, 2) a novel function of microglia in general that is beyond the already described text book knowledge on microglia during brain development that consists of their function for neurogenesis, synaptic pruning, neuronal nutrition and others. Thus, the topic tackled by the authors if of great importance and some of their findings are of big importance.

We are glad the reviewer sees importance in our findings.

Despite the relevance of the topic there are still some weaknesses of the study. Some of their statements are not justified and simply not supported by the data. Further, the flow of the manuscripts should also be improved.

We have reworked the manuscript flow and hope it is now easier to read.

Major concerns:

1. The main message of the paper that CD11c+ microglia are important for myelinogenesis needs to be supported by further data. I strongly suggest to perform myelin thickness measurements (G-ratio)

on myelinating axons by electron microscopy in CD11cCre IGF1fl/fl animals. This important experiment will further strengthen their main message.

As suggested by the reviewer we have now performed EM analysis. It shows significant differences in mean myelin G-ratio and distribution of thinner myelin profiles between the CD11cCre IGF1fl/fl and control mice. Our analysis shows that the less myelinated fibers (G-ratios of 0.8 - 0.85) were found to be over-represented in the CD11cCre IGF1fl/fl animals, the sufficiently myelinated fibers (G-ratio values 0.65 - 0.7) were less represented among the total number of G-ratio values which supports our other data (shown now in Fig 2). We also show that the brains of Tg mice weighed significantly less than controls.

2. Figure 1: The authors state that their CD11b+CD11c+CD45high cells are microglia that is very bold based only on the FACS data. In fact, the expression of CCR2 in this cell population is untypical for microglia and usually restricted to circulating cells. This surface marker profile could also indicate other cells such as monocytes, dendritic cells and activated CNS macrophages. This, the authors should be more careful on their wording until this point. They provide confirmation for the microglia nature of these cells later in Fig. 3 that should therefore be moved already to figure 1.

Apparently the reviewer misread Fig1. We provided flow cytometry data showing that CD11c+ microglia are CD45 dim CD11b+ and in addition they are CCR2 negative and histology showing that they express CX3CR1 and IBA1, which are well-recognized characteristics for microglia.

3. The authors state several times in the text (eg. abstract, title page 9 etc.) that CD11c+ microglia have "stem cell-like potential". However, there is simply no experimental evidence for this strong statement because only transcriptional data are shown. Similarly, the role of this microglia subset was described as "key role in....neurogenesis" (see title) but no evidence is provided for this. Their statement that CD11c+ microglia "are self-renewing" (page 11) is not supported by any data and should be omitted.

We agree with the reviewer that having only transcriptional data is not sufficient to draw such strong conclusions, and we have now corrected these in the manuscript as well as changed the title.

6. It is illogical and not clear why CD11c+ microglia cannot be depleted by their DTR approach in CD11cCre DTR mice. The lack of depletion points to technical difficulties.

The reviewer makes a very good point. We cannot exclude that technical aspects contributed to our findings, for which reason we have modulated our interpretation that this reflects a significant role for these cells. However, we would point out that the only studies we can find that report depletion of neonatal microglia achieved at best 50% depletion using pan-microglial strategies, whereas we have targeted a specific subset. Quite distinct from the response of adult microglia, we note a selective upregulation of itgax with no effect on igf1. This response was exactly similarly induced by all 3 toxin regimens. We feel that this is of interest in itself so even though we cannot ascribe mechanism and because we do not have time for deeper analysis we have chosen to leave these results in the paper, but not interpret them beyond their description.

Minor points:

1. First sentence in the introduction. Please add references for microglia origin, namely Ginhoux et al. Science (2010, already in the reference list) Schulz C et al. Science (2012), Kierdorf et al. Nat Neurosci (2013). The last two references were missing at all.

The references are now added to the manuscript

2. Figure 2: Labelling of the pictures in incorrect. "Tg" should be changed to the correct genotype CD11cCre IGF1fl/fl.

The figure labeling is now corrected accordingly

Referee #2:

In this well-written manuscript by Wlodarczyk and colleagues, the authors study the role of CD11c-expressing microglia in myelination and provide transcriptomic analyses of this potentially distinct microglia subset. They nicely demonstrate that more microglia are CD11c+ during development and, based on previous work, after EAE. They conclude that loss of IGF1 in CD11c+ microglia causes a decrease in myelin protein expression and that there are several cassettes of genes that are expressed more highly by CD11c+ but not CD11c- microglia. If successful, this work is of great potential interest, given the recent implications of microglia in synapse elimination, neuronal survival, and roles in human disease. Indeed, the idea of microglia heterogeneity is a tantalizing one. There remain several major concerns, however, that if addressed would greatly increase the impact of their findings and better validate their conclusions.

1) CD11c+ cells as a distinct population of microglia. The authors nicely demonstrate that there are pockets of CD11c+ cells throughout the developing, and in the injured CNS. It remains unclear to this reviewer: are all microglia somewhat CD11c+, with some small variability in expression? The FACS plots of developing microglia don't show as nice of a clear CD11c+ population as this lab has shown in the group's previous work in EAE. The selected immunostaining images give the impression (I'm guessing falsely) that all microglia are CD11c+. Some additional images could be helpful to clarify. Regarding the issue of if these cells are distinct: the sequencing and qPCR data throughout and specifically for the CD11c+ versus CD11c- microglia require some orthogonal methods for validation, such as proof of good isolation (microglia-specific versus glia or neuronal gene expression) and RNA in situ validation of some differences (in particular true GFAP expression in CD11c+ microglia). It was difficult from the FC data provided what the absolute expression values were for sequencing - especially since other astrocyte-, oligodendrocyte-, and neuron-specific genes were also upregulated in CD11c+ cells according to the provided Supplemental Table 1, implying the possibility of differential cell-type contamination instead of the differential expression of non-microglial genes such as Fabp7, Neurod1, and GFAP.

Our FACS data showed that around 20% of total microglia are CD11c+. We have now provided an overview histology picture to clearly demonstrate that not all of the microglia are CD11c+ and that they are not homogeneously distributed within the brain (Fig1).

Microglia populations were sorted using FACS with purity mode which gave at least 98% purity. We used tight gating around CD45med/dim CD11b+ population and we used isotype controls to discriminate CD11c+ and CD11c- microglia populations. In the revised manuscript we have provided a flow cytometry dotplot showing the isotype control for anti-CD11c (Fig.1).

This method of microglia purification is commonly used for downstream assays such as RNAseq (Butovsky, 2014) (Matcovitch-Natan, 2016).

We validated GFAP expression in microglia by histology stainings (very few GFAP+ CX3CR1+ cells were found in the neonatal brains) and Nestin expression by histology and flow cytometry Fig. 5

Further analysis of absolute expression values from RNA-seq showed us that some of the genes from the heat map provided in Fig. 5 are expressed at relatively low levels, so we have stepped back on our statement about their stem-cell like potential.

2) CD11c+ microglia-derived IGF1 responsible for myelination. PLP immunostaining, while one marker of myelin protein expression, is not necessarily a great way to show decreased myelination. Further, there is no mechanistic insight - either through survey of the literature about IGF1's role in myelination nor about how CD11c+ microglia versus others might play this role. A) Is the decrease in PLP and other myelin protein sufficient to demonstrate a defect in myelination? How does it compare with the IGF1 full null mouse? Or in a mouse with IGF1 loss in all microglia? B) Is the effect via survival of the oligodendrocytes, precursors, or perhaps the neuronal survival effect on projection neurons? Either the author's conclusions should be stepped back or they should provide some mechanistic insight. At least, the authors should show the A) effect on myelination either through another method (EM, physiology, et)? B) # of CD11c-Cre-GFP+ cells in the brain (with staining to show this Cre targets only a subset of microglia and C) Verify the loss of IGF1 only in CD11c+ cells. They show global levels, but other drivers, such as CX3CR1-Cre have been show to have some off-target recombination in neurons developmentally, and given that the authors claim these CD11c+ microglia as a "novel" population, more description of this mouse is helpful and warranted. This could be accomplished by crossing in a recombination reporter. These would also be helpful for the depletion studies, as it would help distinguish robust escaper cells from cells that were impervious to DTA depletion.

As in response to reviewer 1, we performed EM analysis that showed a significantly higher mean myelin G-ratio and higher frequency of less myelinated axons in CD11cCre IGF1fl/fl animals than CTR littermates, which supports our data. We also added additional qPCR data showing decreased Mag gene expression. We have provided flow cytometry analysis showing Cre expression in CD11c+ microglia but not in the CD11c- microglia or CD45 negative cells, moreover we show that IGF1 expression is clearly reduced in Cre+ CD11c+ microglia reaching levels as low as in CD11c- microglia and splenic CD11c+ cells. We also show data that Cre recombination in CD11c+ microglia is around 40%. All data mentioned above are now shown in Fig 2. We feel that these new data make our point much more strongly than initially. We have attempted in the graphical abstract as well as in the discussion to convey the message from this, that CD11c+ microglia play a key role as tissue macrophages controlling myelinogenesis in the neonatal brain.

In addition to these major points, there are several smaller points:

1) Please include n used; on occasion a representative image of an experiment is shown from 3 individual experiments, but not reported as such. It is not always clear what counts as N either, especially for sequencing.

Included in the revised manuscript

2) Absolute counts would be helpful addition to supplemental table 1.

Now included in Suppl 1

3) Are CD11c+ cells more proliferative after injury or depletion? Given their robust appearance after, it could be that proliferating cells are more likely CD11c high?

We previously published that both populations of microglia proliferated equivalently during neuroinflammation (Włodarczyk et al, 2015). During development both populations of microglia expressed genes associated with proliferation and cell division. Moreover, after microglial ablation both populations were found in the repopulating clusters. Taken together we would argue that CD11c is unlikely to be a marker for microglial proliferation

Referee #3:

This is a rather descriptive study with no clear convincing data as well as clear overstatements/over-interpretation of the data presented. The conclusion that the CD11c + population is a myelinogenic and neurogenic phenotype with stem cell-like potential is clearly an overstatement. The data presented here a just correlative and based mostly on gene expression profile. In addition, the conclusion that "Inability to deplete CD11c+ microglia from the developing brain further underlines their importance in neurodevelopment" is not accurate. The authors did not well characterize the depletion specificity and efficiency of their model. The only conclusion possible from their data is that there are not able experimentally to deplete neonatal CD11c+ microglia.

As in response to reviewer 1 we agree that having only transcriptional data is not sufficient to draw some of the strong conclusions, and we have now corrected these in the manuscript. Nevertheless, our study includes functional experiments demonstrating importance of CD11c+ microglia-derived IGF1 for primary myelination. Our revised manuscript includes data that further support this conclusion.

Below are some comments.

The flow cytometry data are not presented at the highest publication standards. The authors need to present the flow plots in biexponential display as well as provide a better CD11c staining with appropriate control. In this regard, the CD11c staining is not resolute! The authors need to show an isotype control to determine where the CD11c population is really starting....as the staining looks rather as a continuum than a define population.

We used matched isotype controls to discriminate CD11c+ from CD11c-microglia both on flow cytometry and FACS sorting. Now the better quality flow cytometry data and the dotplot showing isotype control are displayed in Fig.1

In the Figure 1 immunohistostainings, the CD11c+ population appears not as homogeneously distributed as the usual microglia population. Can the authors document this better? Could the CD11c+ population be in fact perivascular macrophages? Interestingly, the cells are not ramified at all.

Indeed, during postnatal development we observe region specific localization of CD11c⁺ microglia while CD11c⁻ microglia is distributed quite homogeneously. We observe that in regions such as corpus callosum and cerebellar white matter the majority of microglia cells are CD11c⁺ unlike in the other areas of the brain. We now included low power images in Fig.1 to better illustrate this point. We confirmed that CD11c⁺ microglia that we see are localized within the brain parenchyma and not in the perivascular space by including laminin staining (now included in fig1). The amoeboid non-ramified morphology of microglia in neonatal brain has been reported consistent with the cells being highly proliferating and migratory.

Figure 2: mislabeling of the figure legend. There is no k, l and m panels.

The figure legend is now corrected, sorry about that.

Selective ablation of IGF1 under CD11c⁺ cells is not convincing. The conclusion "selective depletion from CD11c⁺ microglia leads to impairment of primary myelination." is not correct as the authors did not formally show that IGF was selectively ablated only in CD11c microglia in these experiments (what about other CD11c⁺ cells?). They need to sort the cells and track carefully recombination. The graph of the quantification of PLP staining intensity is also misleading as the Y axis starts at 70, suggesting a minimal effect.

We now showed by flow cytometry that Cre-GFP is expressed by CD11c⁺ microglia but not expressed by CD45 negative cells, or CD11c⁻ microglia (now included in Fig 3). We mentioned in the text that the efficiency of Cre recombination in CD11c⁺ microglia was at most 40% now we include this information in Fig 3. We sorted Cre- CD11c⁺ MG, Cre+ CD11c⁺ MG, CD11c⁻ MG as well as splenic CD11c⁺ cells from WT and CD11cre IGF1fl/fl and assessed IGF1 expression. We showed that in WT CD11c⁺ microglia the IGF1 expression was much stronger than in other cell populations sorted (Fig 3). The graph with PLP quantification is now presented with a scale from 0. The reduction of Myelin associated gene expression and PLP staining intensity is statistically significant. Furthermore EM analysis showed higher myelin G-ratio and higher frequency of less myelinated axons in CD11cCre IGF1fl/fl animals than in control littermates. We believe that altogether the data clearly show the importance of CD11c⁺ microglia for primary myelination.

The Nestin observation is interesting but is not related to the CD11c⁺ vs CD11c⁻ microglia study as both population express the same level of Nestin. Same for the repopulating experiments, the authors show that some repopulating microglia are CD11c⁺ but do not provide any quantification.

Nestin expression by microglia is of increasing interest for the neuroimmunology community, as evidenced by papers we have cited, thus we feel that our observations showing transient Nestin expression in microglia in developing brain will be of interest even though they do not reflect a subset-specific characteristic. We prefer to show these data but if the reviewer feels strongly we can remove them.

Again concluding that "Taken together, this inability to deplete CD11c⁺ microglia using three distinct approaches underlines their critical importance for development of the CNS" is not possible. There is simply no depletion.

As in response to Reviewer 1, we cannot exclude that technical aspects contributed to our findings, for which reason we have modulated our interpretation that this reflects a significant role for these cells. However, we would point out that the only studies we can find that report depletion of neonatal microglia achieved at best 50% depletion using pan-microglial strategies, whereas we have targeted a specific subset. Quite distinct from the response of adult microglia, we note a selective upregulation of itgax with no effect on igf1. This response was exactly similarly induced by all 3 toxin regimens. We feel that this is of interest in itself so even though we cannot ascribe mechanism and because we do not have time for deeper analysis we have chosen to leave these results in the paper, but not interpret them beyond their description.

About the fact that some clusters of repopulating microglia contained CD11c⁺ cells, until the authors show that these cells are related to the CD11c⁺ cells that they are studying in development, nothing can be concluded. They could be totally unrelated cells that just upregulate CD11c expression due environmental cues.

We observe and had published that re-populating microglia, similarly to neonatal microglia, express nestin as well as show high representation of CD11c⁺ cells. Taking that into account we now asked

whether re-populating microglia recapitulate the neonatal microglia genetic profile. We found that re-populating microglia were different from neonatal, further supporting the differences between neonatal and adult microglia.

About a role of microglia during development, the authors should cite Squarzzoni et al, Cell Reports.

It is now cited

Additional Correspondence

23 June 2017

Thank you for submitting your revised manuscript to The EMBO journal. I am sorry for the delay in getting back to you with a decision, but I have now received the input from the three referees.

As you can see below, the revision received a mixed response. While referee #1 is satisfied with the revised version, referee #3 is not convinced that the present dataset is conclusive enough. Referee #2 is more supportive, but also find that it is not clear how specific the CD11c deletion of IGF1 is and if other cellular subsets are affected by the Cre line. The referee has a number of suggestions for how to address this issue.

Before taking a decision I would like to get your input on what you can do to address this question and the timeline for this. I see that the optimum experiment would be to cross the Cre line to a reporter and look at the expression, but such an experiment will also take much time unless of course you have the mice already. So I am open to other experiments that would address this question.

I would therefore appreciate if you could send me a point-by-point response to the concerns raised by referee #2 that outlines experiments you can do to address the concerns raised.

REFEREE REPORTS

Referee #1:

The authors addressed all main concerns and performed several new experiments that significantly improved the quality of the study. I have no points left and I'm satisfied with this revision.

Referee #2:

Wlodarczyk and colleagues have provided a revision which better illustrates the potential role for CD11c-expressing microglia in myelination, as well as a transcriptomic characterization of a potentially distinct microglial subset. The authors nicely addressed many of this reviewer's comments and concerns, including more definitive proof of myelination defects with IGF1 loss with EM studies and characterization of CD11c cells. One major concern remains: given that the main conclusion of the paper hinges on CD11c-specific manipulation, how specific is the CD11c-specific deletion of IGF1 in the brain? Despite their letter stating that Figure 2 includes "flow cytometry histogram showing Cre recombination selectively in CD11c+ microglia", this is untrue. The authors added data which examined Cre-GFP expression in FACS isolated cells (2C) and examined knockdown of IGF1 in CD11c positive cells (2D). Unfortunately, FACS analysis of neurons, astrocytes and even oligodendrocytes can be difficult, as the cells are extremely fragile (and thus may not be present in the CD45 negative fractions shown). Regardless, Cre expression at a given time does not necessarily relate to recombination, especially in constitutive Cre lines. Given that this Cre has not been well characterized in the brain, the authors should still demonstrate that this previously observed phenotype of dysmyelination in IGF1 dysregulation is not due to loss of IGF1 by OPCs, neurons, or even astrocytes. Crossing these mice to a reporter, or examining the possible knockdown of IGF1 in these other cells remains critical to verify the author's conclusions (for example, a Figure 2D using similar cells as in Figure 2A would firmly establish if IGF1 is lost non-specifically by other cells and a potential major confounder of the findings). If loss of IGF1 is truly limited to the CD11c+ microglia, that strongly supports the author's very important finding of a role

for CD11c+ microglia in myelination.

Referee #3:

The authors did not address adequately the concerns that we raised. Their findings remain correlative and the models used not convincing for the reasons that we outlined in the first revision. Their claims are clearly overstated compared to the data that are provided that are not fully supporting their conclusions.

2nd Editorial Decision

27 June 2017

Thanks for sending me the point-by-point response. I appreciate the proposed experiments and I do think they will add important support for your findings.

I would therefore like to ask you to proceed as indicated and we can continue to discuss how the revisions are coming along.

2nd Revision - authors' response

13 July 2017

Referee #2:

Wlodarczyk and colleagues have provided a revision which better illustrates the potential role for CD11c-expressing microglia in myelination, as well as a transcriptomic characterization of a potentially distinct microglial subset. The authors nicely addressed many of this reviewer's comments and concerns, including more definitive proof of myelination defects with IGF1 loss with EM studies and characterization of CD11c cells.

One major concern remains: given that the main conclusion of the paper hinges on CD11c-specific manipulation, how specific is the CD11c-specific deletion of IGF1 in the brain?

Response: The reviewer is concerned that the CD11c promoter may direct off-target expression of Cre in cells other than microglia. We feel that our data argue against this affecting our interpretation even if it did occur (see below), but we have done new experiments to try to deal with it.

Despite their letter stating that Figure 2 includes "flow cytometry histogram showing Cre recombination selectively in CD11c+ microglia", this is untrue. The authors added data which examined Cre-GFP expression in FACS isolated cells (2C) and examined knockdown of IGF1 in CD11c positive cells (2D). Unfortunately, FACS analysis of neurons, astrocytes and even oligodendrocytes can be difficult, as the cells are extremely fragile (and thus may not be present in the CD45 negative fractions shown).

Response: We agree and disagree. OPC and astrocytes can be and have been studied by flow cytometry, especially astrocytes. We know that the CD45-negative fraction in our flow cytometric analysis of Cre-GFP expression contains astrocytes, since this is how we obtained them in previous studies. Since astrocytes expressed the second highest level of IGF1 mRNA in Fig 2A, we feel that our flow cytometry data therefore excluded that this source of IGF1 was lost in a Cre transgenic cross, since they are in the Cre negative fraction. However, the reviewer is strictly correct that we had not definitively identified that neurons or OPC do NOT express Cre. We have now done this by using a combination of immunohistochemistry and immunofluorescence to identify GFP-expressing cells in neonatal CNS from CD11cCre-GFP IGF1fl/fl mice. Here we should clarify that the Cre transgenic that we used also expresses GFP and so GFP is a reporter for Cre expression. We have provided now full description of the mouse strain as CD11cCre-GFP IGF1fl/fl in the revised manuscript.

Flow cytometry data in Fig 2D had already confirmed that only CD11c+ but not CD11c- microglia or CD45-negative cells (which would include at least some astrocytes) express Cre. We have now performed IHC/IF-stainings on brain tissue collected from 3 PN4-5 CD11cCre-GFP IGF1 fl/fl mice. Our analysis of brains from 3 separate mice reveals GFP+ cells in cerebellum and corpus callosum, consistent with the immunostaining for CD11c in Figure 1. We used an amplification-antibody based protocol to stain GFP as in previous studies so we would not miss weakly-expressing cells.

Importantly, all of the GFP+ cells in neonatal brain were also positive for CD11b, which excludes the possibility that brain cell populations other than microglia expressed Cre.

The only cells that can delete IGF1 are those expressing Cre so the evidence that we now provide in Fig 2C, that CD11b-negative cells do not express Cre, verifies that non-microglial cells could not have deleted IGF1 and therefore that the reason for reduced IGF1 message can only be that the gene was deleted in Cre-expressing CD11c+ microglia (as was already shown by flow cytometry, now Fig 2D).

Regardless, Cre expression at a given time does not necessarily relate to recombination, especially in constitutive Cre lines. Given that this Cre has not been well characterized in the brain, the authors should still demonstrate that this previously observed phenotype of dysmyelination in IGF1 dysregulation is not due to loss of IGF1 by OPCs, neurons, or even astrocytes.

Response: While it is possible that Cre expression might not necessarily relate to recombination, and it may be true that this CD11c promoter has not been well characterized in brain, nevertheless, lack of Cre expression in a cell lineage does eliminate the possibility that the floxed gene was deleted in those cells. We have already verified Cre expression by microglia, as well as reduced expression of IGF1 by these cells. Our new IHC/IF analysis demonstrates that non-microglial (eg CD11b-negative) cells do not express Cre-GFP, whereas we have now shown by two approaches that microglia do. This excludes the possibility that Cre-driven IGF1 gene deletion occurred in neurons, astrocytes or OPC.

Crossing these mice to a reporter, or examining the possible knockdown of IGF1 in these other cells remains critical to verify the author's conclusions (for example, a Figure 2D using similar cells as in Figure 2A would firmly establish if IGF1 is lost non-specifically by other cells and a potential major confounder of the findings).

Response: The reviewer suggests applying the approach in Fig 2A to Cre expression. Reference to Fig 2A is interesting, because if you look at that figure, and sum the expression by neurons, OPC and astrocytes, they come to maybe a third of the expression by microglia, so even total abrogation of IGF1 expression by neurons, OPC and astrocytes, if that should happen, could not have accounted for the over two thirds reduction that we show at 21d (compare 2A with (original) 2F). Our FACS data already provides indirect evidence that astrocytes do not express Cre. The most plausible interpretation for all of this, even without direct data, is that deletion of IGF1 from CD11c+ microglia was responsible for the myelin defects that we describe. However, we now add data that directly show lack of expression by CD11b-negative cells. We referred already to the fact that the Cre transgenic that we used also expresses GFP and so is a reporter for Cre expression.

If loss of IGF1 is truly limited to the CD11c+ microglia, that strongly supports the author's very important finding of a role for CD11c+ microglia in myelination.

Response: To examine the knockdown of IGF1 in specific cell types we have already used FACS-sorted microglia populations and shown significant reduction in mRNA levels in CD11c+ CD45^{low} CD11b+ microglia. Our new data validate that only myeloid lineage cells expressed Cre-GFP, therefore excluding non-microglial cells. We have now added a new Fig 2C to show GFP and CD11b double-stainings in corpus callosum of one neonatal mouse. We also provide for the reviewer's benefit a montage of stainings from corpus callosum and cerebellum of three neonatal CD11cCre-GFP IGF1 fl/fl mice. Loss of IGF1 was therefore limited to the CD11c+ microglia, confirming a key role for this microglial subset in myelination.

We hope these new data with accompanying revisions to the manuscript satisfy the reviewer's concerns and that our paper can be accepted.

3rd Editorial Decision

28 July 2017

Thanks for submitting your revised manuscript and for carrying out the immunohistochemistry analysis on the Cre-GFP mice.

I asked referee #2 to re-review the manuscript. As you can see below, the referee is not convinced that this is a good enough control and that what you really need to do is to measure recombination events and not GFP expression in the CRE-GFP mice. So the experiment is to cross the Cre line to a Cre recombination reporter and look at expression of reporter gene using either histology or FACS.

I have also asked further advice on this issue and the advisor is in agreement that this a relevant control.

I do see that taking all your data into account that the findings support your conclusions, but the above issue has been raised from early on and is a valid one. Your manuscript reports really exciting findings and so lets make sure that the relevant control is in even if this will delay things a bit.

I don't know if you have started the cross already - but let me know the timeline.

Let me know if we need to discuss anything further

REFEREE REPORT

Referee #2:

This reviewer continues to assert that showing lack of recombination in non-microglia cells is critical. We disagree that cells that are GFP negative are definitely not recombined as developmental recombination is definitely possible, as described in the CX3CR1-Cre mice when crossed to a recombination reporter. We still believe that this control (showing recombination is limited to CD11c+ microglia) is essential to support the author's conclusions.

Non-microglial cells are possible to study by FACS. This reviewer's comment was intended to highlight that they may not be represented by the CD45 neg cells, as the authors do not prove it-- and the preservation of these cells types can be largely user dependent. Regardless, GFP expression alone is an insufficient control for recombination as it is a Cre only promoter.

We also appreciate that although we've highlighted this concern since our initial review, the authors have not begun crossing these mice and there may be time pressures given the potentially great impact of their findings if true. Luckily there is an alternative strategy the authors may wish to consider. They could isolate the cell types as they do for the IGF1 qPCR (+proving they've isolated those cells with cell type specific markers), and use genomic DNA PCR to show lack of recombination in these other cell types, especially neurons and oligos the two cell types most often implicated in IGF1-mediated decreases in myelination in IGF1 KO studies. We would highly encourage the authors to obtain a recombination reporter mouse for future studies in any Cre mice if the specific line has not been tried in a new system--it's a frequent control for reviewers (and any discerning reader) to ask for since it is likely critical to support the major conclusions of a sound scientific study.

3rd Revision - authors' response

29 August 2017

Referee #2:

This reviewer continues to assert that showing lack of recombination in non-microglia cells is critical. We disagree that cells that are GFP negative are definitely not recombined as developmental recombination is definitely possible, as described in the CX3CR1-Cre mice when crossed to a recombination reporter. We still believe that this control (showing recombination is limited to CD11c+ microglia) is essential to support the author's conclusions.

Non-microglial cells are possible to study by FACS. This reviewer's comment was intended to highlight that they may not be represented by the CD45 neg cells, as the authors do not prove it-- and the preservation of these cells types can be largely user dependent. Regardless, GFP expression alone is an insufficient control for recombination as it is a Cre only promoter.

We also appreciate that although we've highlighted this concern since our initial review, the authors have not begun crossing these mice and there may be time pressures given the potentially great impact of their findings if true. Luckily there is an alternative strategy the authors may wish to

consider. They could isolate the cell types as they do for the IGF1 qPCR (+proving they've isolated those cells with cell type specific markers), and use genomic DNA PCR to show lack of recombination in these other cell types, especially neurons and oligos the two cell types most often implicated in IGF1-mediated decreases in myelination in IGF1 KO studies. We would highly encourage the authors to obtain a recombination reporter mouse for future studies in any Cre mice if the specific line has not been tried in a new system--it's a frequent control for reviewers (and any discerning reader) to ask for since it is likely critical to support the major conclusions of a sound scientific study.

Response

We thank the reviewer for their patience as well as for a helpful suggestion that has saved us a lot of time.

As the reviewer recommended we have now used MACS to sort microglia, astrocytes, oligodendrocyte precursor cells (OPC) and neurons from brains of PN7 mice, as well as CD11c+ dendritic cells from spleens of the same mice. We used PCR as described in (Liu JL, 1998) to analyze recombination. The results are now shown in new Figure 2D and described in the revised manuscript (on p.6). There was clear recombination in DC and microglia, and nothing at all in astrocytes, OPC or neurons. PN7 is during the time of peak primary myelination and is in the same time frame as we had shown differential expression of IGF1 by CD11c+ microglia versus MACS-sorted astrocytes, OPC or neurons.

We validated purity of the MACS-sorted populations by flow cytometry for CD11b, CD45, ACSA-2 and O4, as recommended by Miltenyi. Neurons did not express any of these markers and additionally could be stained in a cytospin with anti-NeuN antibody. Those profiles and stainings are attached below for the reviewer's attention, we do not feel they need to be published.

Flow cytometry assessment of cell population purity after MACS cell isolation : A) ACSA-2+ Astrocytes, B) O4+ OPC, C) CD45-, CD11b-, ACSA-2-, O4- Neurons

In order to confirm purity of the neurons population we performed D) IF staining of anti-NeuN and DAPI.

4th Editorial Decision

30 August 2017

Thanks for submitting the revised manuscript. Your study has now been re-reviewed by referee #2 and the comments are provided below. As you can see, the referee appreciates the added experiment and is supportive of publication here. There is one last issue that I would like to ask you to sort out in a final revision

REFEREE REPORT

Referee #2:

We thank the authors for their additional experiments to verify lack of recombination in non microglia. If they can provide evidence that the unrecombined version is present/detectable in their isolated cells (right now shown as blank gels in spliced, cropped images), perhaps in the form of a genotyping pcr to show presence of pcr-able DNA, this reviewer is convinced and think this manuscript is of great value to the readership of this journal.

4th Revision - authors' response

30 August 2017

Referee #2:

We thank the authors for their additional experiments to verify lack of recombination in non microglia. If they can provide evidence that the unrecombined version is present/detectable in their isolated cells (right now shown as blank gels in spliced, cropped images), perhaps in the form of a genotyping pcr to show presence of pcr-able DNA, this reviewer is convinced and think this manuscript is of great value to the readership of this journal.

Response

We thank the reviewer for their comments, and positive opinion of our paper.

DNA content for all of the samples that were amplified in the PCR in Fig 2D and Figure EV1 were validated by optical density (NanoDrop) as described in the Methods.

To add to our verification that recombination was confined only to microglia, we have now added an Expanded View Figure EV1 that shows PCR for recombined Igf1 in MACS-sorted DC, and astrocytes from PN21 homozygous CD11c^{Cre-GFP} Igf1^{F1/F1} mice, as well as in MACS-sorted microglia, neurons, OPC and astrocytes from PN7 heterozygous CD11c^{Cre-GFP} Igf1^{F1/WT} mice. As for Fig 2D, the PN21 samples show recombination only in DC and microglia, but not in astrocytes (or in DC from Cre-negative mice). The PN7 samples, being heterozygous, also show a wild-type Igf1 band, which verifies that there was PCR-amplifiable DNA in all of the samples. Again, only microglia show a recombined band, whereas none of the other samples do. This confirms that the CD11c promoter selectively and specifically drove Cre expression only in microglia in neonatal brain.

We have added description of these data, with reference to Fig EV1, to the text on P.6 of a revised manuscript.

Corresponding Author Name: Trevor Owens

Manuscript Number: EMBOJ-2016-96056